# Chirality-induced avalanche magnetization of magnetite by an RNA precursor

S. Furkan Ozturk [1] ✉, Deb Kumar Bhowmick[2], Yael Kapon[3], Yutao Sang [2], Anil Kumar[2], Yossi Paltiel [3], Ron Naaman [2] & Dimitar D. Sasselov [4]

Homochirality is a hallmark of life on Earth. To achieve and maintain homochirality within a prebiotic network, the presence of an environmental factor acting as a chiral agent and providing a persistent chiral bias to prebiotic chemistry is highly advantageous. Magnetized surfaces are prebiotically plausible chiral agents due to the chiral-induced spin selectivity (CISS) effect, and they were utilized to attain homochiral ribose-aminooxazoline (RAO), an RNA precursor. However, natural magnetic minerals are typically weakly magnetized, necessitating mechanisms to enhance their magnetization for their use as effective chiral agents. Here, we report the magnetization of magnetic surfaces by crystallizing enantiopure RAO, whereby chiral molecules induce a uniform surface magnetization due to the CISS effect, which spreads across the magnetic surface akin to an avalanche. Chirality-induced avalanche magnetization enables a feedback between chiral molecules and magnetic surfaces, which can amplify a weak magnetization and allow for highly efficient spin-selective processes on magnetic minerals.

The emergence of life can only be understood within the context of its planetary environment. As such, prebiotic chemistry is constrained by environmental conditions[1]. These constraints also dictate the features of life. One such feature is biomolecular homochirality—single-handedness of the molecules of life.

Reaching and maintaining homochirality is crucial for a robust prebiotic network efficiently producing functional polymers. Despite its importance, origin of homochirality has remained elusive to this date and understanding the ways by which the environment can break the chiral symmetry is paramount in elucidating this mystery. In our recent studies, we addressed this problem and proposed that magnetic mineral surfaces can facilitate enantioselective processes and highlighted the possible role of authigenic magnetite ($Fe_3O_4$) sediments on the origin of homochirality[2–5].

Authigenic iron minerals are ubiquitous in ancient lacustrine environments and they play various roles in geochemical processes as inferred from the Curiosity rover data on Gale Crater, Mars[6,7]. Magnetite is one of the most abundant iron minerals present in the

sediments and magnetite-silica rocks are hypothesized to be covering the bottom of a redox-stratified freshwater lake in Gale crater[8]. Authigenic magnetite formation is also suggested as a mechanism to stabilize liquid water accompanied with the production of $H_2$ which can function as a feedstock in the prebiotic synthesis of biomolecules[9]. In addition, due to their ferrimagnetic nature, $Fe_3O_4$ sediments align their magnetic domains while they form as small superparamagnetic particles under the planetary field and carry a statistically uniform chemical remanent magnetization (CRM)[10,11]. With this distinct net magnetization direction, magnetic sediments break the chiral symmetry on a hemisphere scale and can accommodate asymmetric processes due to a phenomenon called chiral-induced spin selectivity (CISS).

The CISS effect establishes a strong coupling mechanism between electron spin and molecular chirality[12–14]. This coupling forces chiral molecules to interact with electrons in a spin-selective manner[15,16]. Likewise, electrons with a well-defined spin alignment to the molecular frame interact with chiral molecules based on their molecular

[1]Department of Physics, Harvard University, Cambridge, MA 02138, USA. [2]Department of Chemical and Biological Physics, Weizmann Institute, Rehovot 76100, Israel. [3]Department of Applied Physics, The Hebrew University of Jerusalem, Jerusalem 91904, Israel. [4]Department of Astronomy, Harvard University, Cambridge, MA 02138, USA. ✉e-mail: sukrufurkanozturk@g.harvard.edu

handedness[17–21]. In our latest work, we reached homochirality utilizing this phenomenon and showed the importance of magnetic surfaces to induce enantioselective processes in a prebiotic network[3].

To reach a homochiral state in a chemical network, a robust mechanism to break the chiral symmetry, inducing an imbalance between two enantiomers, is required. In addition, a persistent amplification of this imbalance has to accompany it[22,23]. Previous studies have studied various ways to induce an initial chiral imbalance, by a chiral symmetry-breaking agent, like parity-violating energy difference[24], selective adsorption on surfaces[25], conglomerate resolution[26,27], circularly polarized light[28], and longitudinally spin-polarized electrons and muons[29,30], to name a few. Several others have focused on the amplification of an induced enantiomeric excess[23,31,32]. A few studies observed spontaneous symmetry-breaking in solution-phase racemizing compounds and amplified the induced imbalance through conglomerate crystallization[33,34]. However, no studies have demonstrated a method to combine a non-destructive chiral agent with a chiral amplification scheme to achieve a persistent, system-level homochirality—particularly for prebiotically relevant chemistry.

In our latest work, we have demonstrated that, due to the CISS effect magnetic surfaces can act as templates for the enantioselective crystallization of an RNA precursor. Moreover, we have shown that, conglomerate crystallization can accompany the chiral symmetry breaking by the magnetic surface as a simultaneous and well-matched amplification mechanism. By combining these two necessary features to reach homochirality, we obtained enantiopure crystals of ribose-aminooxazoline (RAO) on magnetite surfaces, from a fully racemic solution of RAO[3]. RAO is a pyrimidine ribonucleotide precursor that can be synthesized by the reaction of glyceraldehyde and 2-aminooxazole with high-yields under prebiotically plausible conditions[35,36]. It is a poorly water-soluble compound that is stable against racemization and forming homochiral crystals in the non-centrosymmetric $P2_12_12_1$ space group[35,37,38].

Last but not least, the emergence of homochirality at the stage of RAO can allow for the efficient transfer of homochirality through RNA to peptides due to the high stereoselectivity in the attachment of L-amino acids to tRNA analogs composed of D-ribonucleotides[4,39]. However, it is important to acknowledge that, based on current evidence, RAO serves as a precursor solely to RNA pyrimidines. The prebiotically plausible synthesis of purine ribonucleotides, however, remains an unresolved challenge. If a route to RNA purines from RAO is also discovered, the biological homochirality problem can be simplified by focusing on the production of a single shared RNA precursor with uniform chirality. In such a scenario, once homochirality is achieved in RAO, it can be effectively transferred from RNA to peptides, and eventually, through enantioselective catalysis, to metabolites—therefore to the entire prebiotic network[4]. With these features, RAO is a promising prebiotic compound which can play a central role in the emergence of biological homochirality.

Our previous results on spin-selective crystallization of RAO on Fe$_3$O$_4$ surfaces demonstrate that it is possible to obtain enantiopure RAO from a racemic mixture by a process controlled only by the environment[3]. However, the interaction behind this process is not one-sided: just as magnetic surfaces induce enantioselective processes among chiral molecules, chiral molecules as well can induce spin polarization on magnetic surfaces. In our latest paper, we proposed a model in which the chirality-induced magnetization phenomenon is reinforcing the statistically significant but non-uniform natural magnetization of the authigenic sediments. A small enantiomeric imbalance induced by the natural magnetization of the magnetic surface can be amplified by subsequent dissolution and re-crystallization cycles. During the crystallization process, nearly pure conglomerate crystals cover the magnetic surface and align the magnetic domains underneath them along their chiral molecular axis—enhancing the magnetization of the magnetic surface due to a cooperative feedback (Fig. 1).

In this work, we have experimentally verified this phenomenon (Fig. 2) and showed that previously not magnetized Fe$_3$O$_4$ surfaces can be magnetized by the crystallization of enantiopure RAO for which the magnetization direction is determined by the handedness of RAO (Fig. 3). In addition, we showed that local magnetization by chiral molecules can trigger an avalanche magnetization process and eventually magnetize an area larger than the one covered by the crystals (Fig. 4). Moreover, the area magnetized by the RAO crystals showed a higher magnetic coercivity by about 20 times the modern geomagnetic field, proving the persistence of surface magnetization against possible geomagnetic reversals (Fig. 4c). When considered together with our previous results, chirality-induced magnetization phenomenon paves the way for a cooperative feedback between chiral molecules and magnetic surfaces (Fig. 1). With this feedback, weaker natural magnetization can be amplified and subsequent surface processes can be made highly enantioselective and persistent under prebiotic conditions.

## Chirality-induced avalanche magnetization

The strong coupling of electron spin to molecular chirality established by the CISS effect paves the way for a chemistry controlled by electron spin. Due to the spin-selective interaction of chiral molecules with electrons, achiral magnetic surfaces with net spin-polarization can act as chiral agents and trigger highly enantiospecific processes.

The early CISS experiments have shown that electron flow through a chiral monolayer is spin-selective and the handedness of the monolayer dictates the spin state with high efficiency of transmission[15,16]. Another manifestation of the same effect is observed when a chiral molecule approaches a surface and gets transiently polarized. Namely, an induced electric dipole is formed. This charge polarization is due to the dispersive forces between the molecule and the surface causing a transient flow of electron density. This conceptually simple and mundane phenomenon gets interesting in the case of a chiral potential of a chiral molecule. The transient splitting of charge leads to a partial spin polarization, so that one electric pole is associated with one spin direction and the other pole with the opposite spin—as the electron flow through a chiral potential is spin selective due to the CISS effect. Therefore, a transient spin dipole is realized along the molecular axis of a chiral molecule as it approaches a surface, as shown in Fig. 2a.

What if the surface is magnetic? If the surface is magnetic with a net spin alignment at its surface, then, it couples with the transient unpaired spin of the chiral molecule via spin-exchange interaction. This coupling is a result of the Pauli exclusion principle and it favors the singlet-like state ($E(\uparrow\downarrow)$) with opposite spins. The triplet-like state ($E(\uparrow\uparrow)$) is penalized and the energy difference between these two states is called the exchange energy (or the exchange integral). The exchange interaction is a short-range interaction (few Angstrom scale, relies on the wavefunction overlap), yet a strong one ($\approx 0.01–1$ eV) typically much stronger than room temperature, $k_BT$[40,41]. Therefore, a magnetized surface favorably interacts with a certain handedness of a chiral molecule and breaks the chiral symmetry by doing so. This is the mechanism by which a magnetized surface acts as a chiral seed for selective adsorption and crystallization from a racemic solution, as we have previously observed (See Fig. 1a in refs. 3,17,21,42).

Now if the surface is magnetic, yet, without a net spin alignment, the same interaction can be utilized to magnetize a surface by adsorbing chiral molecules from their enantiopure solution. The fundamental physics underlying the process is identical to the previous case. However, now, instead of aligned surface spins selecting a molecular chirality; molecular chirality is selectively aligning surface spins along the molecular axis, as illustrated in Fig. 2a. Therefore, as an enantiopure layer of a chiral molecule is covering the magnetic domains of a surface, previously misaligned domains align their spins along the same direction, resulting in a net magnetization at the

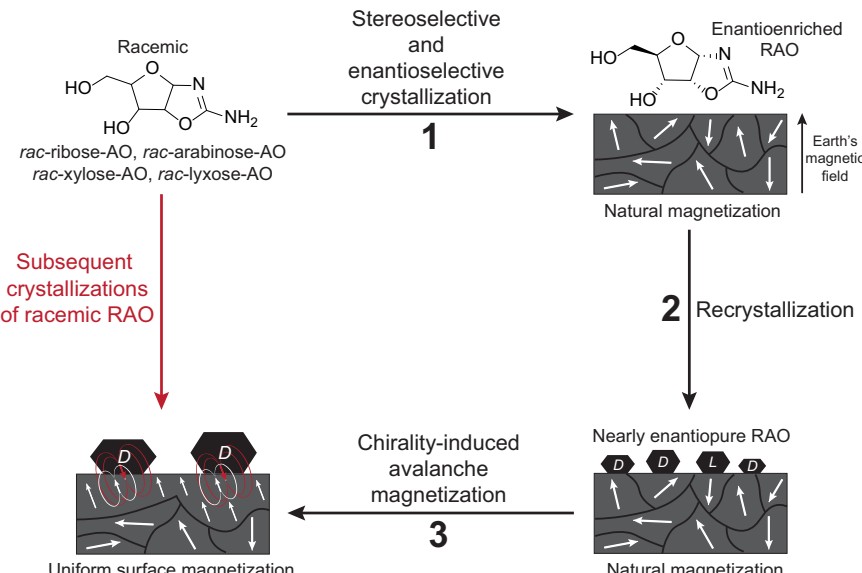

**Fig. 1 | A cooperative feedback between the magnetic surface and RAO can amplify the natural magnetization.** Authigenic magnetic minerals get magnetized while they form under a geomagnetic field. Sedimentary rock surfaces with these magnetic inclusions carry a net remanent magnetization. This natural net magnetization is not uniform albeit statistically significant over the scale of a hemisphere and can allow for spin-selective asymmetric processes due to the CISS effect. In our previous work[3], we have shown that an essential RNA precursor, ribose-aminooxazoline (RAO), can selectively crystallize from a racemic mixture of pentose aminooxazolines on a magnetized magnetite ($Fe_3O_4$) surface (**1**). Although, with a subsequent re-crystallization, we obtained nearly enantiopure crystals of RAO (**2**); in a natural environment, on a surface with non-uniform magnetization, this process will inevitably be less selective. However, the interaction between the magnetic surface and chiral molecules is reciprocal: chiral molecules can magnetize magnetic surfaces due to the spin-exchange and magnetic dipolar interactions. Hence, as enantiopure crystals of RAO form on the magnetic surface, the chirality-induced avalanche magnetization allows for obtaining surfaces with uniform magnetization (**3**). These uniformly magnetized surfaces can be employed again for successive crystallizations of the incoming racemic material (red arrow), with significantly enhanced enantioselectivity compared to surfaces magnetized under the geomagnetic field. Therefore, the self-reinforcing cooperative feedback between the magnetic surface and chiral molecules can enhance the natural magnetization and set the stage for highly selective asymmetric processes on early Earth.

surface. This transition is shown in Fig. 2c, from sub-panels 1 to 2. Previous experiments have shown that chiral molecules can manipulate the magnetization of magnetic surfaces based on their molecular chirality, due to the spin-exchange interactions. Ben-Dor et al. showed that the formation of a chiral self-assembled monolayer can switch the magnetization direction of a previously magnetized substrate[18]. Follow-up work by Meirzda et al. investigated the dynamics of this system with an in-situ NV magnetometer and demonstrated that the angular alignment of surface magnetic dipoles follow the molecular tilt angle over long time scales. The long-time-scale effect of the induced magnetization provided the evidence that spin-exchange interactions are responsible for the magnetization switching[43].

Molecules approaching the surface couple with the surface electrons, aligning their magnetic moments due to spin-exchange interactions, however, this coupling is a short-ranged one, relying on the overlap of wave-functions. Therefore, as the physio-adsorbed chiral molecular layer grows, the strength of the spin-exchange interaction with the surface electrons decreases rapidly. At longer length-scales magnetic dipole-dipole interactions (also known as dipolar coupling) take over. Magnetic moments localized in the chiral molecular layers generate magnetic dipole fields around them (Fig. 2b) and this dipole field can interact with the magnetic dipoles of the surface. This interaction is always present in a system with local magnetic moments (e.g. ferromagnets), however, it is typically ignored due to its weakness compared to the exchange interactions[44]. The thermally averaged magnitude of dipolar coupling, $E_{dd}$, scale as $\mu^2 d^{-3}$, where $\mu$ is the average magnetic moment and $d$ is the distance between two magnetic moments. Due to this slowly decaying inverse cubic ($d^{-3}$) scaling, dipolar coupling is the dominant energy scale at distances longer than a few nanometers into the chiral layers. It is important to reiterate that the local magnetic moment of the chiral molecular layers is due to their charge polarization being accompanied by a spin polarization. And this charge polarization cannot penetrate into the crystal indefinitely as the organic chiral crystals under consideration are not conductive. So it is misleading to think of the whole crystal as a small magnet. Rather, the magnetic behavior is localized to the multi-layer away from the surface whose strength is gradually decreasing as the charge polarization is vanishing away from the surface. To summarize, there are two different interactions dominant at different length scales: short-ranged and strong spin-exchange interactions dominate directly away from the surface (monolayer scale, microscopic); weaker and long-ranged dipolar coupling dominates farther into the crystal (multilayer scale, mesoscopic).

Both of these interactions are present for a chiral crystal forming on the surface as shown in Fig. 2c. Initially, as a monolayer of molecules is forming on a magnetic surface, molecules align the surface magnetic moments underneath them due to exchange interactions. This alignment initiated by the chiral molecules spreads like an avalanche process over the magnetic domains. As molecules align the regions underneath them nearby regions also align themselves similar to the Ising-type ferromagnetic ordering. For this process, chiral molecules can be modeled as a spatially localized external field ($h_0$) in the two dimensional Ising model[45]. The aligned regions attract chiral molecules faster compared to the randomly oriented domains, due to the associated lower spin-exchange energy. Then, the chiral monolayer spreads around and covers more area on the magnetic surface. Simultaneously, the molecular layers get thicker and macroscopic crystal seeds form. As the layers get thicker, exchange interactions of the higher layers with the surface get weaker and dipolar coupling takes over in this region. The chiral multi-layer with a mesoscopic number of local magnetic moments generates a long-ranged magnetic dipole field and couples with the magnetic domains of the surface. This dipolar coupling together with the exchange interactions dominant at short distances, increase the magnetic coercivity of the magnetic

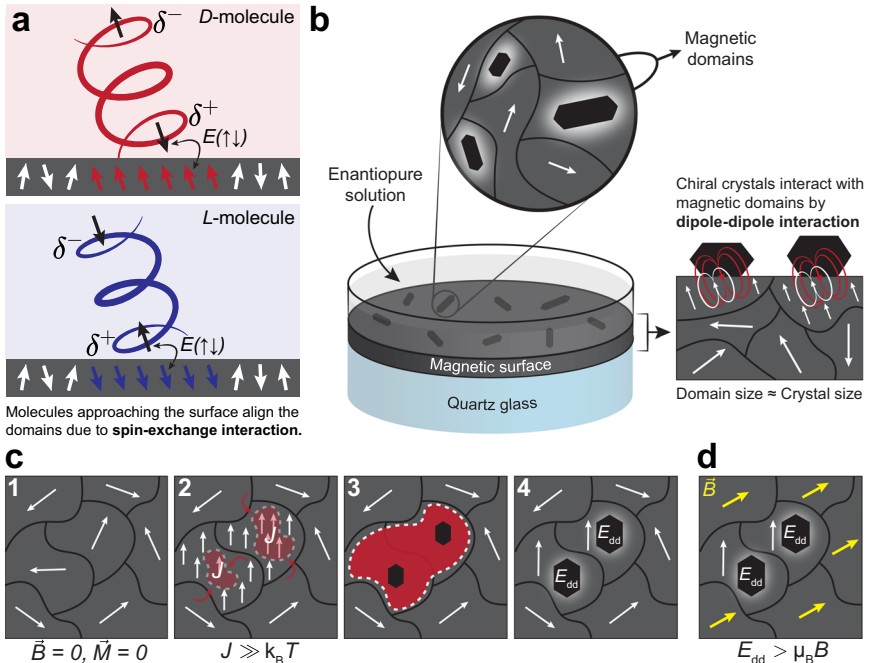

**Fig. 2 | Chiral molecules can spin polarize magnetic surfaces due to the CISS effect. a** Electron density of a molecule approaching a surface is asymmetrically distributed and a transient charge dipole is created. Charge transport through chiral molecules is spin selective due to the CISS effect and therefore a spin-dipole along the chiral molecular axis accompanies the charge dipole. This transient spin-dipole can couple with surface spins due to the spin-exchange interaction ($J \equiv [E(\uparrow\uparrow) - E(\uparrow\downarrow)]$) and spin-polarize the surface along the chiral molecular axis. **b** Schematic of the setup used in crystallization experiments for CD detection. Homochiral crystals of RAO are formed on magnetite from their enantiopure solution. These crystals align the magnetic domains under them and interact with the surface spins due to magnetic dipolar couplinga weaker but longer range coupling compared to the spin-exchange interaction **c** A sequence showing the effect of chiral molecules on the magnetic domains (top view). **1** Initially, there is no net magnetization of the magnetic domains. **2** As a layer of RAO (shaded red area) is forming, the spins underneath the layer align due to the strong spin-exchange interaction. This is an avalanche magnetization process: molecules align the surface spins within a magnetic domain and the aligned regions attract more molecules (red curly arrows) and get larger. **3** The mono-layer grows and covers more surface due to the attraction of chiral molecules to the aligned regions and crystal seeds start forming. **4** Crystals get larger and couple with the magnetic domains due to magnetic dipole-dipole interaction, $E_{dd}$. **d** If an external magnetic field, $\vec{B}$, is applied, the domains outside the area covered by the crystals get magnetized (yellow arrows), yet the crystals preserve the magnetization of the domains under them (white arrows).

surface. Hence, the regions around the chiral crystals preserve their magnetization under external magnetic fields as they require stronger fields to be aligned compared to the free domains without chiral molecules as shown in Fig. 2d.

## Results
### Circular dichroism measurements
We crystallized enantiopure RAO on previously non-magnetized $Fe_3O_4$ surfaces and measured their chirality-induced magnetization by circular dichroism (CD) spectroscopy. (Check Supplementary Information Section 11 for the X-ray crystallographic data of RAO crystals). For these experiments we formed 80-nm-films of $Fe_3O_4$ on 1-mm-thick quartz substrates by electron-beam deposition of iron and its subsequent oxidation to $Fe_3O_4$ as detailed in the Methods section. We then formed *D*- and *L*-RAO crystals, separately, from their 75 mM enantiopure aqueous solutions on magnetite surfaces, as shown in Fig. 2b. Additionally, we repeated the same crystallization experiment on just quartz substrates, without any magnetic sample, as a control experiment to identify the spectral features due to RAO crystals. After an overnight crystallization, we observed an almost uniform coverage of surfaces by small RAO crystals of various sizes in 10–100 $\mu$m range.

Afterwards, magnetite surfaces were probed by CD spectroscopy in the UV-visible range (230–600 nm). During these measurements, the physio-adsorbed crystals were kept on the surface and the samples were placed such that the beam propagation direction is parallel to the surface normal, as shown in Fig. 3c.

Starting with the control measurements we first measured the CD spectrum of enantiopure RAO crystals on quartz such that their contribution to the overall spectrum was isolated (Fig. 3a). The thick red and blue curves in the spectra shown in Fig. 3b were obtained for *D*-RAO and *L*-RAO crystals, respectively. As seen, RAO crystals do not give a CD signal in the visible range and they have a narrow feature around 250 nm. For *D*-RAO crystals the sign of the CD signal is positive and for *L*-RAO it is negative.

Next, we carried out the second control experiment and identified the spectral features of magnetized magnetite for different magnetization directions. For this experiment, a magnetite surface on quartz without any chiral crystals was used. Then an external magnet near the magnetic sample at an angle (about 45 degrees with respect to the surface normal) was placed such that the surface was both in-plane and out-of-plane magnetized (Fig. 3a). Then the CD spectra of magnetite magnetized by the north and south poles of the magnet were measured. The thin red and blue curves in the spectra shown in Fig. 3b were obtained for north and south poles, respectively. A broad spectral feature in the UV-visible range with no zero-crossing was observed. For magnetite films magnetized by the north pole (south) a negative (positive) CD signal was measured. The magnitude of the signal is variable with changing distance of the magnet to the surface and with the magnet strength. For the reported experiment, the out-of-plane

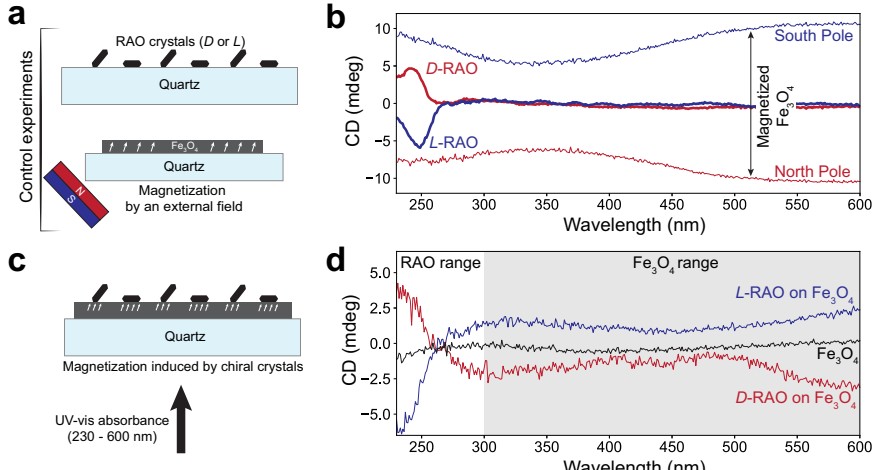

**Fig. 3 | Magnetization of magnetite by RAO crystals is measured by CD spectroscopy. a** We measured the CD spectra of chiral RAO crystals and magnetized magnetite ($Fe_3O_4$) separately, as control experiments. We crystallized *L*- or *D*-RAO on quartz and obtained their CD spectra. We also measured the CD spectra of $Fe_3O_4$ on quartz, magnetized by an external magnet. **b** RAO crystals themselves do not have a significant CD signal in the visible range and they have a narrow peak around 250 nm. *L*-RAO crystals (thick blue curve) give negative and *D*-RAO crystals (thick red curve) give positive CD signal. However, magnetized $Fe_3O_4$ has a broad CD spectrum over the UV-visible range and $Fe_3O_4$ magnetized by north pole (south pole) gives a negative (positive) CD signal. Therefore, in the visible range, the CD signal of magnetized $Fe_3O_4$ is separated from RAO and can be easily distinguished above 300 nm. **c** As our main experiment, we formed enantiopure RAO crystals on $Fe_3O_4$ with no initial net magnetization and measured the induced magnetization by the chiral crystals. **d** As seen in the CD spectra, the initially non-magnetized $Fe_3O_4$ (black curve) becomes magnetized by the adsorption of chiral RAO crystals. *D*-RAO induces north-pole-like, negative magnetization (red curve). Whereas *L*-RAO induces south-pole-like, positive magnetization on $Fe_3O_4$ (blue curve). Note the zero-crossing in the CD signal, around 270 nm, due to the overlap of RAO and $Fe_3O_4$ signals of opposite signs. We refer the reader to Supplementary Fig. 18 for the experimental uncertainties associated with these measurements.

magnetic field strength at the sample location was measured to be around 16 mT.

Finally, as our main experiment, we probed the induced surface magnetization by chiral crystals and measured the CD spectra of magnetite films with adsorbed enantiopure RAO crystals (Fig. 3c). We made sure that the magnetite films used in the experiments were not magnetized prior to the crystal adsorption. As observed in the CD spectra displayed in Fig. 3d, the black line corresponding to the bare magnetite surface is flat, confirming the absence of net magnetization. Yet, as RAO crystals form on magnetite surfaces, they induce net magnetization on the magnetite where the magnetization direction is dictated by the molecular chirality of RAO (See Supplementary Fig. 18 for repeated CD measurements). Changing magnetization direction of $Fe_3O_4$ with changing molecular handedness is a smoking gun proving that the observed phenomenon is due to the CISS effect. As seen in the CD spectra of *L*-RAO (blue line) and *D*-RAO (red line) on $Fe_3O_4$, there are two overlapping features: a narrow feature in the deep-UV range due to enantiopure RAO crystals and a broad feature covering the UV-visible range due to magnetized $Fe_3O_4$. Hence, the spectrum of *D*-RAO (*L*-RAO) on $Fe_3O_4$ is an overlay of two spectra: *D*-RAO (*L*-RAO) on quartz and $Fe_3O_4$ magnetized by the north (south) pole of the magnet. To our delight, this is consistent with our latest work on spin-selective crystallization of RAO on magnetized $Fe_3O_4$, in which we showed that $Fe_3O_4$ magnetized by north (south) pole is promoting the crystallization of *D*-RAO (*L*-RAO)[3]. Thanks to this consistence, a cooperative feedback between the selective crystallization of RAO and chirality-induced magnetization of $Fe_3O_4$ can be conceived.

Although it is hard to measure the strength of the magnetization induced by RAO precisely, comparing the CD signal to magnetite magnetized by an external magnetic field of known strength gives an estimate. With that, we estimate an effective magnetic field of about 2.5 mT (≈50 times the modern geomagnetic field) that would have induced the same magnetization on magnetite as the chirality-induced one by RAO. For this estimate we used the CD angle at 600 nm and the external magnetic field value of 16 mT that was measured at the sample location.

We used solid-state CD spectroscopy to probe the chirality-induced magnetization on magnetite and for these measurements we took advantage of the fact that chiral RAO crystals and magnetized $Fe_3O_4$ give CD signal in different regions of the spectrum with individually recognizable features. Due to the absorptive nature of the measurement we did not exclusively measure the chirality-induced magnetization at the surface, but we also measured the bulk magnetization of the sample. Therefore, the obtained signal is diluted by the bulk $Fe_3O_4$ with no net magnetization. However, using magneto-optical Kerr effect spectroscopy, we exclusively probed the surface magnetization by a reflective measurement and also obtained the magnetic coercivity of the surface upon the adsorption of chiral crystals.

## Magneto-optical Kerr effect microscopy and avalanche magnetization

Magneto-optical Kerr effect (MOKE) describes the change in light polarization reflected from a magnetized surface. MOKE microscopy is an optical imaging technique that relies on the magneto-optical Kerr effect. In the MOKE microscopy, an image is created by the interference of the polarized reference light with the light reflected off a magnetized surface. Therefore, a MOKE microscope image is a polarization contrast image where the bright and dark colors indicate a constructive and destructive interference respectively. A detailed explanation of the technique can be found in refs. 46,47.

We used a commercial MOKE microscope to image the magnetic domains of a nickel surface and probed the effect of chiral crystals on the surface magnetic properties. We first imaged the magnetic domains of the substrate in its demagnetized state and made sure that we could resolve the domains. For the MOKE measurements we could not use the polycrystalline $Fe_3O_4$ surfaces as we used for the CD measurements, due to their small domain size below the optical resolution of the microscope (≈1 μm). We fabricated a 30-nm-thick nickel (Ni) surface covered by a 5 nm gold (Au) layer by electron-beam evaporation for the MOKE measurements. The Au coating reflects the spin property of the underlying magnetic layer and is applied to

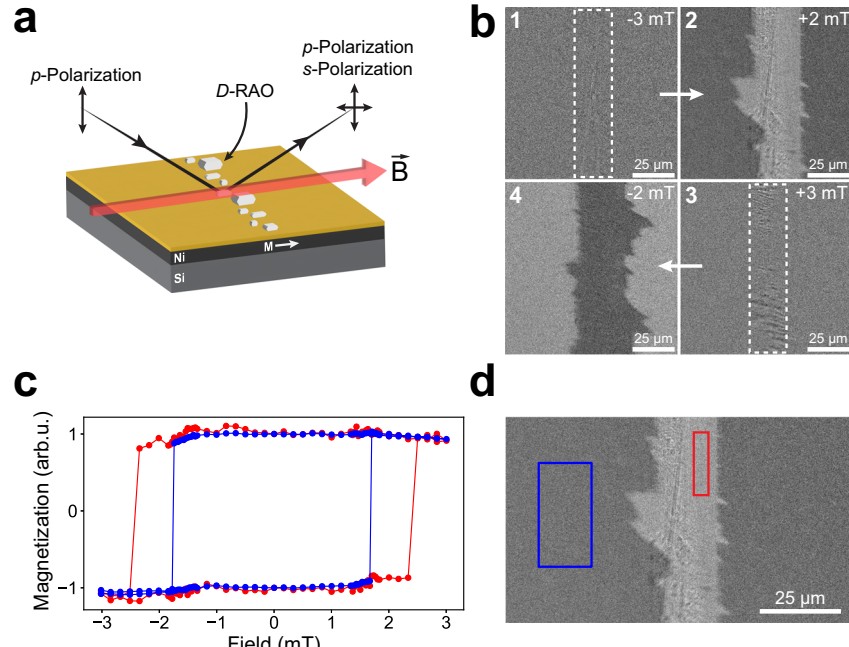

**Fig. 4 | With a magneto-optical Kerr effect (MOKE) microscope, surface magnetic domains are imaged. a** We imaged the magnetic domains of a Nickel/Gold substrate with a MOKE microscope in the longitudinal configuration. We formed the RAO crystals on the surface as a thin line and imaged the domains surrounding the crystals. **b** Kerr microscope images of the magnetic domains with the *D*-RAO crystals are shown. White dashed lines surround the area where the crystals are formed, as deduced from the optical images. Each image here shows a snapshot of the domains as an external, in-plane magnetic field is swept from −3 mT to +3 mT and then back to −3 mT. **1** All of the domains are aligned by the strong and negative magnetic field of −3 mT. **2** As the magnetic field is swept from −3 mT to +2 mT, the free domains outside of the regions affected by the crystals flip their magnetization. However, the regions nearby the chiral crystals do not flip and this creates the image contrast. These regions require a more positive field to flip their

magnetization due to the extra effective magnetic field locally induced by the chiral crystals. **3** A more positive magnetic field of +3 mT is able to flip the domains around the crystals and align all of the magnetic domains. **4** Lastly, as the magnetic is swept back from +3 mT to −2 mT, the free domains align themselves under the magnetic field and the domains nearby the crystals do not flip until a more negative field is applied. **c** The Kerr hysteresis loop of the free domains (blue-boxed region in **d**), sufficiently away from the chiral crystals have a coercive field of around 1.5 mT (blue points). In contrast, domains near the RAO crystals (red-boxed region in **d**) experience an effective magnetic field from spin-exchange and magnetic dipolar interactions, resulting in a higher coercive field of around 2.5 mT (red points). **d** MOKE image of the magnetic surface when a 2 mT in-plane magnetic field is applied.

prevent the oxidation of Ni under ambient conditions[48]. After demagnetizing the sample, we measured its in-plane domain size to be around 10 μm (Supplementary Fig. 20), suitable for the MOKE measurements. We did not optimize the thickness of the ferromagnetic layer or test a multi-layer sample topology to reduce the out-of-plane magnetic anisotropy[49]. We obtained the highest MOKE contrast by imaging the in-plane magnetic domains and we observed a sharp phase transition from a demagnetized state to a saturated one as we gradually increased the external longitudinal magnetic field.

Because the MOKE measurements rely on an optical image of the magnetic surface, scattering from the crystals affects the image and it is not always possible to image directly underneath the crystals with high accuracy. Therefore, we formed small crystals to minimize the scattering and analyzed the magnetic domains nearby the crystals. Before every MOKE measurement we first took a magnetically saturated reference image at a fixed light polarization, which was then subtracted from each subsequent MOKE image (See "Methods").

Having studied the bare sample under the MOKE microscope, we formed enantiopure RAO crystals on the Ni/Au surface and observed a drastic change in the magnetic properties. We formed the crystals either by spin-coating or drop casting a 20 mM aqueous solution of enantiopure RAO. The crystals were about 1-10 μm and they formed a dense layer on the magnetic surface. For the spin-coating experiments, a thin stripe of RAO crystals were formed upon spinning the solution, as seen in Fig. 4. We also formed small crystals by rapidly cooling the solution of drop-casted *L*- and *D*-RAO and made sure that we obtained similar results with spin-coating methods (Supplementary Fig. 25).

After forming the crystals on the Ni/Au surface, we imaged the magnetic domains at room temperature, under external magnetic field in the longitudinal configuration for which the in-plane magnetization is parallel to the reflection surface, as seen in Fig. 4a. We applied the in-plane magnetic field with switchable polarity using a pair of electromagnets built around the imaging plane of the microscope. For the MOKE measurements, we first saturated the magnetic surface by applying a strong magnetic field of −3 mT and took the reference image (1 in Fig. 4b). We then started to sweep the magnetic field to the opposite direction, from −3 mT to 3 mT. During this sweep, free domains far away from the crystals flipped their magnetization at about 1.5 mT, yet, the domains around the crystals kept their negative magnetization until the field is raised to 2.5 mT. This gives rise to a MOKE contrast between the free domains and the ones near the crystals (2 in Fig. 4b). Then, by applying 3 mT field we saturated the domains again but now in the opposite direction (3 in Fig. 4b). Afterwards, we swept the magnetic field back in the more negative direction and we observed a similar behavior. At around −1.5 mT field the free domains flipped their magnetization and the domains around the crystals did not until −2.5 mT, resulting in a similar MOKE contrast (4 in Fig. 4b).

As a control, we fabricated a sample surface with a magnetic (Ni/Au) and non-magnetic (Si) areas adjacent to each other and we formed the *D*-RAO crystals on this surface. We then imaged the intersection under the MOKE microscope and observed the effect of chiral crystals only on the magnetic side of the surface (Supplementary Fig. 27). This measurement confirms that the observed effect is not an imaging

artifact and is due to a magnetization contrast. Furthermore, we repeated the MOKE measurements with achiral compounds, sodium chloride (NaCl) and glycine, and confirmed that the achiral crystals do not interact with the magnetic surface (Supplementary Fig. 25 and Supplementary Movie 4). In addition, MOKE measurements with L-alpha-helix polyalanine molecules were performed to further verify the spin-exchange origin of the chirality-induced magnetization phenomenon using a different chiral molecule (Supplemntary Fig. 28).

We also measured the Kerr hysteresis loop of the magnetic domains far from and near the RAO crystals and observed a significant change in the coercivity of the magnetic domains due to the chiral crystals. This can be explained by an effective magnetic field influencing the domains nearby the chiral crystals due to the spin-exchange and magnetic dipolar interactions. As shown in Fig. 4c, we recorded the MOKE contrast change during an in-plane magnetic field sweep for two regions: a region far from the area covered by the crystals (blue box in Fig. 4d) and a region nearby the crystals (red box in Fig. 4d). Hysteresis measurements (Fig. 4c, Supplementary Fig. 25, and Supplementary Movies 1–3) show that chiral crystals significantly increase the coercivity (magnetic resistance) of the magnetic surface by about 20 times the modern geomagnetic field[50], although the magnetic domains we could analyze were not directly under the crystals. The prebiotic importance of this is two-fold. First, the presence of chiral crystals make the surface magnetically harder, preserving the surface magnetization under higher external demagnetizing fields than Earth's geomagnetic field. Second, the magnetization is obtained and retained on an area larger than the one covered by the crystals. Therefore, the magnetization triggered by the chiral crystals is non-linear and spreads like an avalanche. This avalanche magnetization allow for a feedback between the magnetic surface and RAO crystals upon which higher surface magnetization can accompany higher enantiomeric excess (Fig. 1).

## Spin-polarization properties of RAO

Lastly, the spin-polarization properties of *L*- and *D*-RAO were measured with a magnetic conductive atomic force microscope (mc-AFM). The

mc-AFM setup consists of a non-magnetic AFM tip measuring the current through the adsorbed molecules on the magnetic substrate as a function of the applied electric potential and the magnetization direction of the substrate, as shown in Fig. 5a. The magnetization of the substrate is switched by an external magnetic field.

With the mc-AFM setup, we measured the spin-polarized current passing through a chiral layer of RAO molecules as a function of the applied voltage across the chiral layer. We measured the current ($I$) for an up ($I_\uparrow$) and down ($I_\downarrow$) magnetized surface for each enantiomer, where up ($\uparrow$) and down ($\downarrow$) refer to the direction of the north magnetic pole relative to the adsorbed layer. Thereby, we measured the difference between the transport efficiency of up and down spin polarized electrons through L- and D- molecules as a direct measurement of the CISS properties of RAO.

As the substrate, we used the same Ni/Au surfaces used in the MOKE measurements due to the high conductivity of gold. Magnetite surfaces are not ideal for the mc-AFM measurements as magnetite is not a good conductor at room temperature. We then prepared RAO multi-layers on Ni/Au surfaces by spin-coating.

For the mc-AFM measurements, we used the contact mode and kept the AFM tip grounded while scanning the substrate voltage from −2 V to +2 V. We simultaneously measured the current flowing perpendicular to the substrate through the chiral multi-layer and obtained the averaged voltage-current (*I-V*) curves in Fig. 5b, c.

As seen in the averaged *I-V* curves in Fig. 5b, c, electron transfer through RAO layers is spin-selective. In the positive voltage regime, up-magnetized current ($I_\uparrow$, north) is more efficiently transferred through the L and down-magnetized ($I_\downarrow$, south) current is more efficiently transferred through the D enantiomer. Therefore, chiral molecules introduce a spin-selective resistivity to the system.

We can calculate the relative spin-polarization of RAO multi-layers by subtracting the up and down-spin currents in the non-linear region (voltage above the charge injection threshold) of the *I-V* curves and dividing by the total current: $\left(\frac{I_\uparrow - I_\downarrow}{I_\uparrow + I_\downarrow}\right) \times 100$. As shown in Fig. 5d, *L*-RAO (blue) and *D*-RAO (red) have approximately equal and opposite

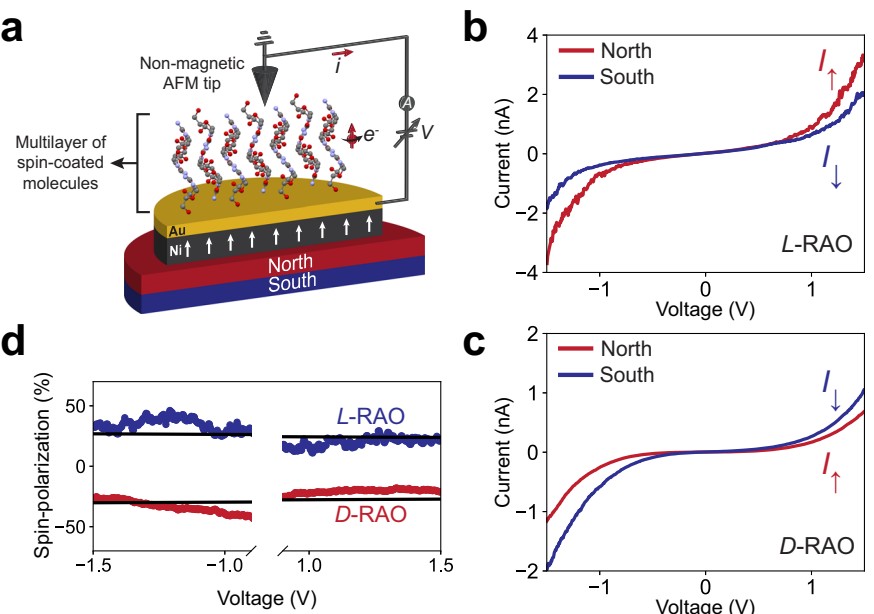

**Fig. 5 | Spin filtering effect of RAO is measured by magnetic-conductive atomic force microscopy (mc-AFM). a** Schematic of the mc-AFM setup. We formed a multilayer of enantiopure *L*- or *D*-RAO on a magnetized Nickel/Gold substrate by spin-coating. **b**, **c** Averaged current-voltage (*I-V*) curves of *L*-RAO (**b**) and *D*-RAO (**c**) are shown. The red (blue) curves correspond to the current measured while the magnet was pointing upwards/north (downwards/south). **d** Percent spin-polarization of *L*- and *D*-RAO are calculated from the non-linear region of the averaged *I–V* curves by $\left(\frac{I_\uparrow - I_\downarrow}{I_\uparrow + I_\downarrow}\right) \times 100$. Black lines are linear fits to the curves whose ordinates are used to extract the spin-polarization. *L*-RAO has a positive spin-polarization of 25%, while *D*-RAO has a negative spin-polarization of −28%.

spin-polarization of about ± 25%, extracted from the linear fits (black lines) to the data in the non-linear region of the *I-V* curves.

We should mention that this is a lower bound to the intrinsic spin-polarization of RAO as the probed area by the AFM tip does not only contain an orderly arrangement of RAO molecules but likely an ensemble of quasi-orderly packed layers. Therefore, the measured spin-polarization is averaged over this quasi-orderly arrangement of RAO. In principle, *I-V* curves could be measured over a crystal of RAO and a much higher spin-polarization could be achieved. However, because RAO crystals are not conductive, the thickness of the crystal has to be small enough such that a current can be measured.

## Discussion

### Implications on prebiotic chemistry

A prebiotic proto metabolism relies on the prebiotic synthesis of homochiral nucleotides until a full transition to nascent biology is achieved. Therefore, achieving homochirality once may not be sufficient for a prebiotic network and the persistence of attained homochirality may be required. A transient and/or spontaneous chiral bias may induce enantiomeric excess at a point in prebiotic chemistry. However, this does not guarantee the maintenance of homochirality in a persistent basis. A solution to this problem is to maintain the same chiral bias in the environment and keep reinforcing the previously attained enantiomeric excess in a deterministic fashion. Therefore, the average magnetization state of the magnetic surface is preserved in the environment, ensuring the persistence of homochirality. The chirality-induced avalanche magnetization phenomenon helps to achieve this in multiple ways. First, by amplifying the natural, authigenic magnetization it sets the stage for spin-selective processes (e.g., crystallization, reduction) with higher degrees of enantioselectivity. Second, it spreads the uniform surface magnetization in a non-linear manner due to the avalanche effect and helps to realize spin-selective phenomena on a larger scale. Finally, it increases the magnetic resistance (coercivity) of the surface by about 20 times the modern geomagnetic field. Of course, this added coercivity due to the chiral molecules will diminish as the molecules are washed away. Nevertheless, the chirality-induced surface magnetization will persist even after the molecules are removed and it will be protected by the bare coercivity of the magnetic surface, which is about 30 times the modern geomagnetic field for our surfaces. Although the Hadean geodynamic record is controversial, the combination of early Archean measurements and modeling indicates that early Earth's magnetic field strength was weaker than the modern one by about a factor of 4[50]. This further ensures that the same chiral bias due to the uniform surface magnetization can be present in the environment despite possible magnetic field fluctuations and geomagnetic reversals.

We proposed that magnetic surfaces could serve as a strong chiral bias, based on the enantiospecific interaction between chiral molecules and magnetic substrates—as presented when the CISS effect was studied. This is possible due to the robustness of chirality-induced avalanche magnetization. Like other CISS-based phenomena, it relies on (≈0.01−1 eV) spin-exchange interactions instead of much weaker (a few nano-eV per Gauss) magnetic interactions. Therefore, the effect can strongly manifest itself under ambient conditions and in natural environments.

Chirality-induced avalanche magnetization phenomenon is a mechanism to enhance the natural magnetization of magnetic rocks. By crystallizing chiral molecules on magnetic minerals, surface spins can be aligned, and these minerals can then accommodate efficient spin-selective asymmetric processes on early Earth. In our previous work, we used magnetic surfaces to break the chiral symmetry and obtained homochiral crystals of RAO from racemic starting materials. To fully manifest the effect in experimental timescales, we used strong external magnetic fields and uniformly magnetized the substrates. However, applying strong magnetic fields does not mimic early Earth

conditions. As demonstrated here, chirality-induced avalanche magnetization is a prebiotically plausible and robust mechanism to obtain a uniformly magnetized surface. Therefore, we do not rely on the presence of an unnatural external field to facilitate spin-selective, asymmetric processes. However, a small natural magnetic field is still helpful as a uniform source of hemisphere-scale symmetry-breaking. Authigenic magnetization of magnetic minerals under the geomagnetic field can initiate the spin-selective processes and guarantee a common direction on a large scale. Our suggested mechanism can utilize a small natural asymmetry in the distribution of magnetic domains and enhance it to a large scale and uniformity. Without this common trigger, different mesoscopic locations on a lake may end up enriching different enantiomers in a stochastic way.

Magnetite is the most commonly found magnetic mineral on Earth. Moreover, it is the most magnetic natural magnetic mineral[10]. Authigenic magnetite sediments form and magnetize under the geomagnetic field in a wide range of depositional environments[9,10]. Such depositional environments are well-preserved in the ancient Mars record, such as the Gale crater[6–8]. In these environments, magnetite is typically found as inclusions in silica-based rocks and can be highly polarized as single-domain particles[8,51,52]. For those reasons, magnetite is likely the most suitable natural magnetic mineral to accommodate the mechanism we consider.

### Modified ising model of the chirality-induced magnetization

As is well established, the standard Ising model is a simplified model of a ferromagnet, and it accounts for the formation of magnetic domains —distinct regions with uniform magnetization[53,54]. The formation and collective alignment of magnetic domains is the primary reason why the magnetization induced by the chiral molecules spreads like an avalanche, in a non-linear fashion. In addition, the size of the magnetic domains compared to the size of the chiral crystals is an important parameter. As such, there should be enough chiral molecules to flip sufficient number of spins in a domain to induce domain flipping.

In any ferromagnetic material, spin-exchange interaction aligns the nearby dipoles by forcing them to point in the same direction, resulting in a magnetic domain. However, as the system contains more aligned dipoles in a larger domain, it generates a large magnetic field around itself and stores high magnetostatic energy. In order to minimize its internal energy, the ferromagnet splits into smaller magnetic domains with opposite dipolar alignment—reducing the surrounding magnetic field. Therefore, the balance between spin-exchange and magnetostatic energies at a given temperature dictate the size of magnetic domains. In a realistic ferromagnet with a crystal lattice the system is more complicated as it takes more energy to magnetize the material in a certain direction (called the easy-axis) compared to the others. Therefore, the formation and size of domains are further balanced by this factor called the magnetocrystalline anisotropy energy.

In our case, we suggest adding another exchange force to the standard ferromagnetic exchange interaction. This exchange force between the chiral molecules and the ferromagnetic material is strong and localized[18]. Therefore, although the real ferromagnetic materials and chiral molecular systems are complicated, avalanche magnetization of a magnetic surface by the chiral molecules can be simply modeled by a two-dimensional Ising-like Hamiltonian, $H$:

$$H = -h_0 \sum_i S_i - J \sum_{\langle ij \rangle} S_i S_j - J_c \sum_{\langle ik \rangle} S_i \sigma_k \qquad (1)$$

where $S_i$ is the spin of the magnetic surface at a given site $i$, $\sigma$ is the transient spin vector localized on an electric pole of a chiral molecule, $h_0$ is an external magnetic field, $J$ is the spin-exchange energy of the magnetic surface, and $J_c$ is the spin-exchange interaction energy between the surface and chiral molecular spins. The first two terms of

this Hamiltonian are the standard Ising model and the third term accounts for the interaction between the magnetic surface and chiral molecules. In the limit of frozen chiral molecular spins, the third term of the Hamiltonian simplifies to a sum over only the surface spins: $J_c N_{nb} \sigma \sum_i S_i$, where $N_{nb}$ is the number of neighboring chiral molecules interacting with a given surface spin. Therefore, the external magnetic field affecting the surface spins is replaced by an effective field in the presence of chiral molecules: $h_0 \rightarrow h_0 + (N_{nb}\sigma)J_c$. This field can be considered as an effective magnetic field due to the chiral molecules. However, due to its spin-exchange origins, it is much stronger ($\approx 0.01$–$1$ eV) and shorter-ranged (a few nanometers) than a regular magnetic field. The two-dimensional square-lattice Ising model is a simple but accurate approximation of our system, and it predicts a second-order phase transition below the critical temperature ($k_B T_c / J \approx 2.269$)[55]. As can be seen in Supplementary Fig. 37, the formation of magnetic domains at a temperature above the critical temperature results in a surface with no net magnetization. However, upon the addition of chiral molecules, the domains under and nearby the chiral molecules start flipping to a common direction and the surface carries a net magnetization whose direction is dictated by the molecular chirality. At lower temperatures, the spread of magnetization should be more drastic due to the formation of larger magnetic domains. However, the CISS-induced spin-exchange interaction energy ($J_c$) of chiral molecules also decreases at higher temperatures resulting in a less efficient domain flipping. The trade-off between these two should be explored with further experimentation.

## Methods

### Materials
Reagents and solvents were obtained from Sigma-Aldrich, Thermo Fisher, Acros Organics, and Santa Cruz Biotechnology. They were used without further purification, unless otherwise specified. A Mettler Toledo SevenEasy pH Meter S20 combined with a Thermo Fisher Scientific Orion 8103BN Ross semi-micro pH electrode was used to measure and adjust the pH to the desired value. All of the experiments were performed under ambient conditions, unless otherwise specified.

### Method for the synthesis and characterization of RAO
*D*- and *L*- ribo-aminooxazolines were synthesized by the reaction of two equivalents of cyanamide (5 g, 0.12 mol) with one equivalent of the *D*- and *L*- ribose (9 g, 0.06 mol) in aqueous ammonia (3.5%, 10 mL). After the reaction, enantiopure RAO was crystallized, the solution was filtered, the crystals were dried and grounded into powder. Racemic RAO was made by mixing *D*- and *L*-RAO powders in equal amounts. Subsequently, the CD spectra of *DL*-RAO was checked to confirm that it is racemic.

RAO crystals were dissolved in deuterium oxide, and their $^1$H-Nuclear magnetic resonance (NMR) spectra were acquired using a Bruker Ultrashield 400 Plus or Bruker Ascend 400 operating at 400.13 MHz. The notations s, d, t, and m represent the multiplicities singlet, doublet, triplet, and multiplet signal, respectively. Chemical shifts ($\delta$) are shown in ppm.

*D*-RAO (yield: 85%): $^1$H-NMR (400 MHz, Deuterium Oxide) $\delta$ 5.73 (d, $J = 5.1$ Hz, 1H), 4.91 (t, $J = 5.3$ Hz, 1H), 4.05 (dd, $J = 9.6$, 5.5 Hz, 1H), 3.86 (dd, $J = 12.7$, 2.4 Hz, 1H), 3.66 (dd, $J = 12.7$, 4.8 Hz, 1H), 3.54 (ddd, $J = 9.6$, 4.7, 2.4 Hz, 1H).

*L*-RAO (yield: 84%): $^1$H-NMR (400 MHz, Deuterium Oxide) $\delta$ 5.72 (d, $J = 5.0$ Hz, 1H), 4.90 (t, $J = 5.3$ Hz, 1H), 4.05 (dd, $J = 9.5$, 5.5 Hz, 1H), 3.85 (dd, $J = 12.7$, 2.4 Hz, 1H), 3.65 (dd, $J = 12.7$, 4.8 Hz, 1H), 3.54 (ddd, $J = 9.5$, 4.7, 2.4 Hz, 1H).

### Method for fabricating magnetic surfaces
Magnetite ($Fe_3O_4$) surfaces were fabricated according to the procedure of Jubb and Alan[56] by evaporating a 40 nm layer of iron on a 1-mm-thick quartz glass (Ted Pella, INC., PN: 26012) using electron beam evaporation under a high vacuum of $5 \times 10^{-6}$ Torr. After the iron layer formation, the samples were baked at 175 °C for 4 h in the air and the oxidation of iron (Fe) to magnetite ($Fe_3O_4$) was promoted. As a result, semi-transparent magnetite samples suitable for CD measurements were obtained.

For MOKE and mc-AFM measurements ferromagnetic Ti/Ni/Au surfaces were fabricated by electron-beam evaporation under a high-vacuum of of $3 \times 10^{-7}$ Torr. The surfaces were prepared by evaporating 8 nm Ti on a 0.5 mm thick silicon (100) wafer followed by 30 nm Ni and 5 nm Au layers. The thin gold coating was used to prevent the oxidation of the magnetic substrate and preserve its spin-polarization properties.

All of the surfaces were cleaned thoroughly with acetone and subsequently in ethanol before every experiment.

### Method for the characterization of magnetite surfaces
**FTIR spectroscopy.** The vibrational spectrum of the magnetite films was characterized using Fourier transform infrared (FTIR) spectroscopy. For the FTIR measurements, a KBr window from ThorLabs (25 mm diameter, 5 mm thick, WG10255) was used, chosen for its low absorption in the mid-IR range. The KBr window was coated with a 20 nm iron film via e-beam evaporation and subsequently oxidized to achieve a 40 nm thick $Fe_3O_4$ film. A commercial FTIR spectrometer by Bruker (Model: Invenio, 8000 cm$^{-1}$ to 350 cm$^{-1}$) was used for the measurements.

A resolution of 5 cm$^{-1}$, a sample scan average of 100 scans, and a phase resolution of 16 using the Mertz phase correction mode were employed for the measurements. The light source utilized was MIR, with a KBr beam-splitter and an RT-DLaTGS detector. Measurements were conducted within the range of 700–350 cm$^{-1}$. A KBr window was placed in the background arm, and the magnetite sample surface was placed in the sample arm of the spectrometer, and the background subtraction feature of the spectrometer was utilized. The measurement revealed a prominent peak, centered at 557 cm$^{-1}$, as shown in Supplementary Fig. 8. The peak's central position aligns closely with the observed phonon mode splitting of magnetite, as previously documented by Jubb and Allen (560 cm$^{-1}$)[56]. Spectroscopic features of maghemite (439 and 546 cm$^{-1}$) and hematite (436 and 526 cm$^{-1}$) were not observed. The FTIR spectra elucidated that the iron films underwent selective oxidation into magnetite, as opposed to other iron oxide phases like maghemite and hematite.

**UV-Vis spectroscopy.** The optical spectrum of the magnetite films was characterized using UV-Vis spectroscopy. UV-Vis measurements were conducted using an uncoated, broadband UV fused silica window by ThorLabs (25.4 mm diameter, 1 mm thick, Part number: WG41050), chosen for its low absorption in the optical range. The fused silica window was coated with a 20 nm iron film, resulting in a 40 nm magnetite film upon oxidation. A commercial UV-Vis spectrometer by Shimadzu (Model: UV-1900, 190 nm to 1100 nm) was utilized. The measurements were taken in the 190–700 nm range with a resolution of 2 nm, employing the slow acquisition mode. In the background arm, a fused silica window was placed, while in the sample arm, the magnetic sample was positioned. The background-subtracted UV-Vis spectrum was obtained. A broad absorption in the UV-Vis range, characteristic of magnetite, was observed, as depicted in Supplementary Fig. 9.

**SQUID measurements.** The magnetic properties of the magnetite samples were characterized using a superconducting quantum interference device (SQUID). Magnetic measurements of oxide films were conducted using an MPMS3 SQUID magnetometer (LOT-Quantum Design Inc.) by applying a vibrating sample mode. The magnetization of the samples was measured with the magnetic field aligned parallel to the surface normal (out-of-plane). The samples were positioned in a

plastic straw. To center the sample position, a small magnetic field was applied, and the so-called lastscan measurement was taken.

The SQUID measurements on the silicon substrate utilized 200-nm-thick magnetite samples. The silicon substrate was separately measured, and its diamagnetic contribution was subtracted. The hysteresis curve of the sample was measured with a magnetizing run from 0 Oe to 10 kOe, a first measurement from 10 kOe to −10 kOe, and a second one back to 10 kOe, concluding with a demagnetizing run from 10 kOe to 0 Oe. A scan rate of 50 Oe s$^{-1}$ was employed. The coercive field was measured to be around 600 Oe, confirming the soft magnetic nature of magnetite. Furthermore, it was observed that the sample was not saturated even at fields as high as 10 kOe, as depicted in Supplementary Fig. 10.

## Method for crystallization experiments

For the CD experiments, 75 mM solutions of *D*- or *L*-RAO were prepared in 2 mL pure water. Afterwards, magnetite surfaces were placed horizontally in a polystyrene Petri dish (35 mm by 10 mm) and filled it with the RAO solution such that the surfaces were fully covered with the solution. The experiments were left overnight at room temperature for crystallization. After the crystallization was completed, the mother liquid was slowly filtered out without disturbing the crystals on the surface. Then the crystals were let to dry on the magnetite surface. After the surfaces were dried, their solid state CD spectra were measured. During these processes, it was made sure that the surfaces did not come in contact with a strong magnet which would contaminate the chirality-induced magnetization.

For the MOKE experiments, 20 mM solutions of NaCl, glycine, and *D*-, *L*-, or racemic RAO were prepared in pure water. The crystals were produced by two different methods. In the first method, one side of the magnetic sample was covered with a tape such that at the intersection of the covered and uncovered sites, a thin stripe of RAO crystals can form upon spinning the solution, as seen in Fig. 4. With this approach, a 1 mL solution of RAO was drop casted on the magnetic surface, which was followed by spin-coating (Laurel Technologies Spin Coater WS Series) the sample. The sample was left then to dry overnight and eventually the tape was removed. The crystals were formed at the tape borders as thin lines. In the second method, small (around 1–10 um) and dense crystals were obtained by drop casting 5 µL of *L/D/*racemic RAO or NaCl or glycine on Ti/Ni/Au (8/30/5 nm) surfaces. Samples were then placed in the fridge (−18 °C) for two cycles of 2 h, with a 2 h interval at room temperature. Finally, samples were dried overnight at room temperature and areas of amorphous aggregation and dense crystallization were obtained.

## Method for CD measurements

Circular-dichroism measurements were taken by using a Chirascan VX CD spectrometer (Applied Photophysics) equipped with a thermoelectrically controlled (Quantum Northwest TC125) single-cell holder. The sample chamber temperature was constantly monitored and kept stable at 20 °C by an active temperature control. A baseline spectrum of the bare quartz substrate was measured before the measurements, revealing no significant CD signal. Substrates were held upright in a cuvette holder and placed perpendicular to the beam direction during the measurements. It was made sure that the substrates fully cover the aperture of the spectrometer. Measurements were taken in the 210–600 nm wavelength range and the auto-subtraction feature was used to subtract the baseline. CD and UV-Vis absorption spectra were simultaneously measured and a step size and bandwidth of 1 nm, and a 1 s time per point were used for the acquisition.

For the control experiment with externally magnetized magnetite, a neodymium magnet with a magnetization of 240 mT was placed at a 45 degrees angle and about 1 cm away from the substrate, in the sample holder. The out-of-plane magnetic field strength at the substrate surface was measured to be around 16 mT.

## Method for MOKE measurements

MOKE imaging and magnetometry were performed by a commercial Evico Magnetics GmbH magneto-optical Kerr microscope. The measurements were taken in the longitudinal configuration. An in-plane magnetic field was generated by an electromagnet obtained from the microscope supplier powered by a Kepco BOP 100-4DL power supply. For the optical imaging of substrates, 20 × or 50 × Zeiss objective lenses were used.

For the MOKE imaging, first a magnetically saturated reference image was taken at a given light polarization, which was adjusted to achieve a high image contrast. Active feedback with a piezo controller was used to mechanically stabilize the sample during the measurements. Each MOKE image was then subtracted from the reference measurement and an image contrast due to the surface magnetization was obtained. The sign and absolute value of the MOKE contrast is not a physical magnetic moment value as it corresponds to a photon count change with respect to a reference image. However, for two MOKE images with a common background, the difference in the image contrast is physical and it reflects a difference in the magnetization. The assignment of darker and brighter image colors to negative and positive magnetization values is also arbitrary and brighter (darker) colors to more negatively (positively) magnetized domains were assigned without loss of generality. For the Kerr hysteresis measurements, the optical intensity that reflects the magnetization states was averaged over a selected region and then normalized to the saturation magnetization.

**MOKE measurements with α-helix polyalanine molecules.** The magnetization induced by α-helix polyalanine (AHPAL) molecules adsorbed on magnetic nickel-gold surfaces was studied to verify the generality of the chirality-induced magnetization phenomenon beyond RAO crystals.

*L*-α-helix polyalanine molecules ([H]-CAAAAKAAAAKAAAA-KAAAAKAAAAKAAAAKAAAAKAAAAK-[OH]), with C denoting cysteine, A representing alanine, and K standing for lysine, were purchased and utilized in the experiments. The experiments involved the preparation of a 1 mM solution of the molecules in ethanol, which was subsequently drop-casted onto the magnetic substrate. A 1 µL drop was cast onto a Ti/Ni/Au (8/30/5 nm) surface. The sample was then left in the air at room temperature until complete drying was achieved. Visible aggregates were detected through optical imaging in the region of the drop-cast, and measurement of magnetic domain flipping induced by the chiral molecules was performed using MOKE imaging (Supplementary Fig. 28). The well-studied covalent bonding to the top gold layer of the ferromagnetic layer was ensured by the thiol end group of the α-helix polyalanine.

## Method for mc-AFM measurements

Samples for magnetic-conductive probe atomic force microscopy (mc-AFM) measurements were prepared by spin-coating (Laurel Technologies Spin Coater WS Series) 20 mM aqueous solutions of enantiopure *D*- or *L*-RAO on Ti/Ni/Au (8/30/5 nm) surfaces. Magnetic field dependent current-voltage (*I-V*) characteristics of the prepared samples were determined using a multimodal scanning magnetic probe microscopy (SPM) system equipped with a Beetle Ambient AFM and an electromagnet with R9 electronic controller (RHK Technology). Voltage spectroscopy for the *I-V* measurements was performed by applying voltage ramps with a non-magnetic platinum tip (DPE-XSC11, MikroMasch with a spring constant of 3–5 N m$^{-1}$) in contact mode. During the mc-AFM measurements the magnetic substrate was kept magnetized with an external, out-of-plane magnetic field of 0.5 Tesla.

## Reporting summary

Further information on research design is available in the Nature Portfolio Reporting Summary linked to this article.

## Data availability

The data that support the findings of this study are available from the corresponding author upon request. The X-ray crystallographic coordinates for structures reported in this study have been deposited at the Cambridge Crystallographic Data Centre (CCDC), under deposition numbers 2220348 and 2220349. These data can be obtained free of charge from The Cambridge Crystallographic Data Centre via www.ccdc.cam.ac.uk/data_request/cif.

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

## Acknowledgements
The authors thank John Sutherland and Jack Szostak for helpful discussions, suggestions, and feedback. We thank Ziwei Liu for the synthesis of RAO samples. We also acknowledge other members of the Simons Collaboration on the Origins of Life and the Harvard Origins of Life Initiative for fruitful discussions that shaped the ideas behind this work. This work was supported by a grant from the Simons Foundation 290360 to D.D.S.

## Author contributions
S.F.O., Y.P., R.N., and D.D.S. designed the research; S.F.O. and D.K.B. made the magnetic surfaces and took the CD measurements; S.F.O. and Y.K. did the MOKE experiments; S.F.O. and Y.S. did the SQUID and spin-coating experiments, A.K. took the mc-AFM measurements; S.F.O. analyzed the data; S.F.O. wrote the paper; all authors contributed to the editing of the manuscript and the Supplementary Information.

## Competing interests
The authors declare no competing interests.
