## [Peer Review File · Nature Communications]

Chirality-induced avalanche magnetization of magnetite by an RNA precursorReviewers' Comments:

Reviewer #1:

Remarks to the Author:

Ms. No.: NCOMMS-23-20352

The authors have demonstrated quite clearly, that adsorption of chiral crystals on the surface of a magnetic material can influence the polarity of the underlying magnetic domains. This is an interesting piece of work, and novel with respect to the growing body of work studying the CISS effect at various types of interfaces. It is very well written and readable. I think that it is entirely appropriate for publication in Nature Communications.

I have a couple of questions that should be readily addressable.

1. In the initial experiments, the authors indicate that their samples were thin films of Fe₃O₄ deposited onto a quartz substrate. Do the authors know the chirality of their quartz substrate? Do they have any thoughts on how its chirality might influence any of their results?

2. p. 9, par. 2, ln. 2 affect  affects

3. Have the authors attempted the growth of RAO crystals from a racemic solution phase? If so, where there any signs of net spin polarization in the magnetite substrate?

Reviewer #2:

Remarks to the Author:

The manuscript is logical continuation of research by the authors of the CISS effect and enantioselective processes. In their recent publication authors showed that magnetization of authigenic iron minerals can cause chirality of the ribose-aminooxazoline (RAO) molecules (precursor of RNA) by enantioselectivity. In the manuscript authors continue their study and demonstrate experimentally that also chiral molecules can induce magnetization in previously not magnetized Fe₃O₄ surface. Moreover they show that avalanchlike magnetization beyond area covered with RAO is possible. These findings together with the previous study are suggested as a possible reason of homochirality of life on Earth.

Experimental measurements are complemented with theoretical simulations within the (suggested by authors) modified Ising model. The study is one more step in the exciting research in CISS and enantioselective processes. It is logical and very interesting. Thus, the manuscript definitely deserves to be published.

There are two minor points authors may (optionally) consider before the publication:

1. Idea of homochirality is explained by presence of original small imbalance in magnetization "on a hemisphere scale" which is enhanced via a cycle of magnetization and re-crystalliation described in the manuscript. However, one can argue that there is also the other hemisphere, where original imbalance should be the opposite.

2. Results of theoretical description within the modified Ising model depend crucially on value of the parameter J_c and on assumption of "frozen chiral molecular spins". This raises two questions:

a) How was estimated value of J_c implemented in the calculations?

b) How accurate is assumption of frozen molecular spins. Does it contradict general description of mutual influence between magnetization and chirality?

Reviewer #3:

Remarks to the Author:

In previous work, Ozturk et al. demonstrated how RAO (a prebiotic RNA precursor) can be recrystallised in diastereo- and enantiopure form if a polarised magnetic surface, such as magnetite under the influence of Earth's magnetic field, is present. This may have provided optically pure RAO which could then lead to enantiopure mononucleotides and thus homochiral RNA. The present work investigates and confirms a supposition of the previous paper - that homochiral RAO can exert influence on the spin polarisation of magnetite surfaces and thus determine the chirality of subsequent crystallisation of more RAO creating a positive feedback loop whereupon larger and larger areas would accumulate the same enantiomer of RAO, making homochiral mononucleotide and RNA synthesis more probable.

I find that the work has been explained in depth and thoroughly carried out, with ample experimental detail to allow reproduction and validate the conclusions. I believe this paper will be publishable after the following minor points are addressed.

To facilitate review, I converted the pdf to a word document and added line numbers, which are used below

Abstract, lines 13-14: An environmental factor is not necessarily required to achieve homochirality, as demonstrated by the Soai reaction. Whilst a prebiotic example of this type of auto amplification of chirality has not been demonstrated yet, in theory it could be possible. Please amend.

Abstract, lines 16-17: Following on from above, please correct e.g. "...magnetic surfaces is crucial for this approach.."

1st paragraph, lines 26-27: Obtaining high yields of chemical reactions doesn't necessarily require enantiopure compounds, the emphasis must be on obtaining functional polymers, which is well known.

4th paragraph, lines 60-62: Considering the length of the manuscript, another sentence or two describing briefly how RNA can (potentially) confer chirality to peptides won't go amiss, particularly as this appears to be one of the most salient points of the paper.

Fig. 1. I'm not sure this scheme makes sense. Once avalanche magnetisation has taken place (top left) the arrow points to a magnetite surface with randomly magnetised domains and RAO crystallising on it. Shouldn't the arrow point to a magnetite surface with a uniform surface magnetisation and racemic RAO crystallising onto it in enantiopure form i.e. the need for further recrystallisation (arrow from bottom right to bottom left) is not required to achieve high ee. Also, for the bottom left image, should the text in black not read "nearly enantiopure RAO"?

Fig 3. line 263: "...D-RAO crystals (thick blue curve).."

Some more general points/comments to be addressed:

1. How big are the naturally occurring domains of magnetised Fe₃O₄? The range of typical sizes would be of interest to the reader and help visualise the process.
2. From the MOKE experiments, it is impressive to see the resistance to spin flipping when chiral crystals are present. When the crystals are washed away, however, presumably the magnetic hardness of that area is lost? If so, how does the magnetic resistance compare to that of Earth's magnetic field? We are told in the presence of RAO it is ~ 20 times.
3. Looking at Fig. 2. C, we see that enantiopure RAO, formed elsewhere, can then recrystallise on a randomly magnetised domain and align the spins of that domain and create the avalanche

magnetisation process, which allows further RAO to crystallise in enantiopure form. However, the implication from the diagram is that when the boundary of that domain is reached, the avalanche magnetisation process is halted. Is this correct? If so it should be stated, and if not I would change the diagram.

Some of the experimental detail given in the main text could be moved to the Experimental section or SI. Eg. The majority, if not all, of lines 272-275, 384-404 and 490-498 could be removed.

Reviewer #4:

Remarks to the Author:

This work follows on from a recent publication (ref. 20, arXiv:2303.01394) showing symmetry breaking and chiral amplification for RAO due to the CISS effect on magnetite surfaces. The expansion of that work to the current "avalanche" feedback system is intriguing. The work should be published, but the authors appear to be overstating the case beyond what the data can support.

- RAO is quite a special molecule. The generalization of this effect to other systems is not yet warranted. Have the authors experimented with other conglomerate systems to show that this is not a special case.
- The authors state that "the emergence of homochirality at the stage of RAO allows for the propagation of homochirality through RNA to peptides and therefore to the entire prebiotic network." RAO can only lead to nucleoside C, and the challenge of making other nucleosides in a prebiotically relevant way remains. Likewise, the extension to peptides, which is cited as the "cyanosulfidic network" of Sutherland, is not straightforward, in that these amino acids are produced in racemic form in that network. You'd need to get to RNA itself producing amino acids to make this claim.
- It is curious that the authors cite only a 1953 theoretical paper to discuss "persistent amplification" of an imbalance. The current work is not related to the mathematical model in the Frank paper. The authors have neglected important work in this area: Kondepudi, Blackmond, Videma, Soai.

In summary, until the authors can show generality beyond one special chiral molecule, the paper should dial back on the hype and present the clear science.

Reviewer #5:

Remarks to the Author:

The manuscript by Ozturk et al. investigates the interactions between chiral molecules and the magnetic state of a sample, in particular with respect of the magnetic properties. The title of the manuscript may be flashy, but to my view does not adequately describe the real content of the paper. First, the studies do not only encompass magnetite, but also Ni films. Second, the main message of the manuscript is that the observed effects can all be traced back to chirality-induced spin selectivity (CISS). Other mechanisms are not discussed and leave the impression that CISS is the only mechanism. The claim is that the CISS effect establishes a strong coupling mechanism between electron spin and molecular chirality. There are many groups worldwide addressing this issue, but with mixed success. Therefore, given the controversial understanding and interpretation of CISS I find this view somewhat biased.

The samples investigated in this study are basically created by a liquid phase approach, bringing enantiopure solutions of RAO or LAO onto the surfaces and letting them dry to form crystals at the surface. The templates were magnetite crystals, magnetite thin films and Ni thin films. As the CISS arguments in the manuscript draw heavily on the exchange interaction between magnetic surface and

the molecules, one immediate question arises:

What is the chemical and electronic nature of the contact between the molecules and the magnetic surfaces? The drying process is certainly leaving behind some residue around the molecular crystals and very likely also at the interfaces. It is also known that oxides, such as magnetite, may have a different chemical composition at the surface, which deviates from the bulk one (if the oxygen concentration at the surface is changed, the material may turn from Fe_3O_4 to Fe_2O_3 , which is actually an antiferromagnet). The same is going to happen with magnetite films and the Ni films, which however have been protected by a Au cover layer (5nm). Therefore, in all of the cases, it cannot be assumed that the molecular crystals are in direct contact with a ferromagnet. As a consequence, electronic exchange interaction which requires an overlap of the participating wave functions will not work, but this is argued to be the main mechanism of CISS.

The magnetic data, particularly, the domain imaging experiments look interesting. Apparently, there is an influence of the molecular crystals (D-RAO) on the domain formation and the magnetization behavior of the Ni/Au film structure. Around the line-shaped trace of the D-RAO crystals, the coercivity is enlarged. Can the authors exclude that this is just a consequence of the changed chemistry at the surface/chemistry due to preparation procedure of the molecular crystals? The enlarged coercivity is going to change the magnetic domain structure, particularly, if the max field values are chosen between the switching fields of the D-RAO area and the Ni/Au area. Again, I believe that the distance argument (see 1) eliminates CISS as a likely mechanism for this behavior.

The AFM investigations are interesting, too, but again misleading. The authors talk about a spin-polarization of the RAO molecules, but their experiment never measures a spin-polarization. What they are actually measuring are differences in the $I(V)$ curves by means of a non-magnetic (!) tip as a function of the applied magnetic field direction. Both L-RAO and D-RAO samples indeed exhibit opposite changes with the magnetic field direction. This indicates a magnetic field effect at work in the $I(V)$ curves, but is not a measurement of the spin-polarization. Again, one has to keep in mind that the Ni layer is protected by the 5 nm Au film and the Au/molecule surface has been treated by spin coating. As a result the average distance between Ni and molecules is larger than 5 nm. I don't understand how the exchange interaction claimed for CISS is going to bridge this large distance. In fact, even the authors state the the exchange interaction may be strong, but is very short ranged.

These are my major difficulties with this manuscript. Although it shows some interesting observations, which still await interpretation, I have strong doubts that these can be unanimously explained with the CISS effect. Therefore, I cannot recommend this paper for publication in Nat. Comm.

We thank **Reviewer #1** for their interest in our manuscript and their encouraging feedback. We are pleased to say that their remarks have been helpful in improving our manuscript. We will respond to their remarks in the same order they put them.

1. *In the initial experiments, the authors indicate that their samples were thin films of Fe₃O₄ deposited onto a quartz substrate. Do the authors know the chirality of their quartz substrate? Do they have any thoughts on how its chirality might influence any of their results?*

For the CD experiments achiral fused quartz slides were used. The technical information about the GE 124 Quartz material can be found at the providers website: https://www.tedpella.com/histo_html/GE124-Quartz.aspx. The quartz substrate is optically transparent in the 0.19 to 4.2 μm range, and it does not yield a net CD signal in the regime of interest, proving its achiral nature (Fig. S16 solid black line). Moreover, the CD signal of the Fe₃O₄ films formed on the quartz substrate was measured before the crystallization of the chiral crystals and no net CD signal was obtained from the non-magnetized Fe₃O₄ film. (Extended Data Fig. 1., black curve). These measurements show that the magnetic substrate before the formation of chiral crystals does not have a net magnetization.

2. *p. 9, par. 2, ln. 2 affect  affects*

Necessary changes were made.

3. *Have the authors attempted the growth of RAO crystals from a racemic solution phase? If so, were there any signs of net spin polarization in the magnetite substrate?*

Yes, we have tried the experiment suggested by the reviewer and measured the induced magnetization by CD measurements. Most of the crystallizations from the racemic solution of RAO yielded no net magnetization of the magnetic substrate, as expected.

We thank **Reviewer #2** for their interest in our manuscript and providing highly positive and encouraging feedback. We will respond to their remarks in the same order they put them.

1. *The idea of homochirality is explained by the presence of original small imbalance in magnetization "on a hemisphere scale" which is enhanced via a cycle of magnetization and re-crystallization described in the manuscript. However, one can argue that there is also the other hemisphere, where the original imbalance should be the opposite.*

The reviewer is indeed correct and the symmetry-breaking we suggest is a hemisphere-scale phenomenon. This certainly allows for the fact that life emerging on the planet can be left- or right-handed, depending on the local environment it emerges in. Overwhelming evidence from phylogenetics, evolutionary biology, and biochemistry suggests that life as we know it has a common origin, a unique ancestor commonly referred to as the last universal common ancestor (LUCA). The fact that life has a common, last universal ancestor does not refute the possible presence of preceding common ancestors that are extinct, known as the family of first universal common ancestors (FUCA).

As illustrated in the tree of life above, common ancestors may have originated in different locations on the Earth from a homochiral pool of nucleotides with left- or right-handed genetic code depending on the hemisphere they have originated in. However, all but one of these common ancestors became extinct and only of them managed to successfully form a biosphere and become the LUCA. The fact that life may have originated simultaneously in multiple locations/ponds on Earth is possible but the life we know it seems to have a common and a single origin which either emerged in the northern or southern hemisphere with a homochiral, right-handed genetic code.

2. Results of theoretical description within the modified Ising model depend crucially on value of the parameter J_c and on assumption of "frozen chiral molecular spins". This raises two questions:

a) How was the estimated value of J_c implemented in the calculations?

In our simulations we normalized the energy scales with the exchange energy J and the parameter of choice was J_c/J —the relative strengths of the exchange interaction between the spins of the ferromagnet themselves and the spins of the ferromagnet with the chiral molecules. Given that the chiral molecules are spin-polarized near unity (30% spin polarization measured by the mc-AFM measurements is likely a lower bound due to the quasi-orderly arrangement of the spin-coated molecules) and the chiral molecules are in closer proximity to the surface than the ferromagnet spins themselves, we estimated this parameter to be sufficiently larger than 1. In a ferromagnet, spins are separated by the lattice spacing and the separation is typically about a nm. However, molecules are in direct proximity to the surface as their wavefunctions overlap with the surface. So, we estimate the characteristic distance for J_c to be around an Angstrom. Because the spin-exchange interaction is highly distance dependent, the interaction with the chiral molecular spins (J_c) is likely to be much stronger than the interaction of surface spins within themselves (J). Direct measurements of the exchange interaction between chiral molecules and magnetic substrates also support this claim (Ref 40 and 41). As seen in the simulations below, $J_c/J > 1$ is sufficient to realize a near-uniform domain flipping we

observed in the experiments and for values higher than about 2, the uniformity of the magnetization saturates and does not increase further.

b) How accurate is assumption of frozen molecular spins. Does it contradict the general description of mutual influence between magnetization and chirality?

The assumption of frozen molecular spins is reasonable due to the well-defined chemical link between the molecules and the surface. In fact, a well-defined link aligning molecules from a common point/moiety is necessary to realize a net spin polarization. Each molecule is likely to interact with the oxide surface through their highly polarizable diol moieties and this energetically favorable link prevents molecules to flip randomly and get attached from an opposite end—leading to an opposite spin polarization. If molecules could randomly orient themselves near the surface as they are charged polarized, no net spin polarization would be realized. And this would neither yield a net magnetization of the magnetic surface, nor enantioselectivity in crystallization on a magnetized surface.

We thank the reviewer for their insightful remarks and questions.

We thank **Reviewer #3** for their interest in our manuscript and their encouraging feedback. We are pleased to say that their remarks have been helpful in improving our manuscript. We will respond to their remarks in the same order they put them.

1. *Abstract, lines 13-14: An environmental factor is not necessarily required to achieve homochirality, as demonstrated by the Soai reaction. Whilst a prebiotic example of this type of auto amplification of chirality has not been demonstrated yet, in theory it could be possible. Please amend.*

At the end of the mentioned sentence, “required” was changed to “desirable”. Although for breaking the symmetry, spontaneous chemical amplification is possible, in order to maintain the imbalance persistently an environmental factor is highly beneficial. This ensures that at various stages of prebiotic chemistry, synthesis of chiral molecules can be along a common direction biased by the environmental agent. We thank the reviewer for highlighting this.

2. *Abstract, lines 16-17: Following on from above, please correct e.g. “..magnetic surfaces is crucial for this approach..”*

In this sentence we do not mean that “magnetic surfaces” are universally crucial to break the chiral symmetry. What we mean is simply that due to our previous success in utilizing magnetized surfaces as chiral agents, inducing a net magnetization is crucial. This is implied in the previous sentence.

3. *1st paragraph, lines 26-27: Obtaining high yields of chemical reactions doesn't necessarily require enantiopure compounds, the emphasis must be on obtaining functional polymers, which is well known.*

We changed the sentence to: “Reaching and maintaining homochirality is crucial for a robust prebiotic network **efficiently** producing functional polymers.” We thank the reviewer for the suggestion.

4. *4th paragraph, lines 60-62: Considering the length of the manuscript, another sentence or two describing briefly how RNA can (potentially) confer chirality to peptides won't go amiss, particularly as this appears to be one of the most salient points of the paper.*

We thank the reviewer for their suggestion which we totally agree with. We made the following changes and cited our most recent paper on how homochirality can propagate in a prebiotic network. We also separated this paragraph to isolate and highlight its important content.

“Last but not least, the emergence of homochirality at the stage of RAO allows for the efficient transfer of homochirality through RNA to peptides due to the high stereoselectivity in the attachment of L-amino acids to tRNA analogs composed of D-ribonucleotides. Therefore, the biological homochirality problem can be simplified by focusing on the production of a single shared RNA precursor with uniform chirality. Once homochirality is achieved in RAO, it can be effectively transferred from RNA to peptides, and eventually, through enantioselective catalysis, to metabolites—therefore to the entire prebiotic network. With these features, RAO is an important prebiotic compound which can play a central role in the emergence of biological homochirality.”

5. *Fig. 1. I'm not sure this scheme makes sense. Once avalanche magnetization has taken place (top left) the arrow points to a magnetite surface with randomly magnetized domains and RAO crystallizing on it. Shouldn't the arrow point to a magnetite surface with a uniform surface magnetization and racemic RAO crystallizing onto it in enantiopure form i.e. the need for further recrystallisation (arrow from bottom right to bottom left) is not required to achieve high ee. Also, for the bottom left image, should the text in black not read "nearly enantiopure RAO"?*
caption

We acknowledge the confusion of the reviewer and thank them for pointing it out; however, we illustrated the magnetic substrate as 2D cross sections from the side (front view), and the relevant part of the substrate is its top surface. We decided to do so to display magnetic domains and emphasize that the chirality-induced magnetization phenomenon is a surface effect. Because the illustrations are 2D cross sections, it is not possible to draw the top-left arrow to the very top of magnetic substrate. However, we made clarifying changes in the figure by moving the text to the top and making the “higher selectivity” arrow start from the top. And we also clarified that in Fig. 2., we illustrated the surface as a projection from the top (top view). We changed the text to “nearly enantiopure RAO”. We thank the reviewer for these suggestions.

6. *Fig 3. line 263: “..D-RAO crystals (thick blue curve)..”*

We are afraid we do not understand the confusion here. D-RAO crystals correspond to the thick red curve and L-RAO crystals correspond to the thick blue curve in the panel of Fig.3b.

7. *How big are the naturally occurring domains of magnetized Fe₃O₄? The range of typical sizes would be of interest to the reader and help visualize the process.*

We thank the reviewer for their intriguing question that we are currently investigating for a future work. The domain structure and hence the magnetic properties of natural magnetite is heavily dependent on how it forms in the environment. If magnetite crystallizes from a more concentrated solution of ferrous iron, typically the magnetite crystals become smaller (10-50 nm) and have single-domain structure. If magnetite crystals are larger, they can either be in the pseudo-single domain regime or in the multi-domain regime. The magnetite we are currently investigating is in the pseudo-single regime with a crystal size of around 100 nm. These magnetite crystals are ideal for high surface spin-polarization without being magnetically too soft. As they exist in the pseudo-single domain regime, each crystal can be thought of as a purely spin-polarized unit. On the other hand, the synthetic magnetite substrates we fabricated for the experiment have much larger domains, but, still not resolvable by optical imaging. We estimate their range to be around and less than 1 micron. That's why for the MOKE experiments we employed Ni/Au surfaces with larger domains such that we can 'watch' the domain flipping. Magnetic domains of the Ni/Au substrate before the RAO crystals are adsorbed can be seen in Fig. S19. They are around 10 microns.

8. *From the MOKE experiments, it is impressive to see the resistance to spin flipping when chiral crystals are present. When the crystals are washed away, however, presumably the magnetic hardness of that area is lost? If so, how does the magnetic resistance compare to that of Earth's magnetic field? We are told in the presence of RAO it is ~ 20 times.*

When the crystals are washed away, the added magnetic hardness to the surface is lost. However, the bare magnetic hardness of the surface still remains. As seen in Fig. 4c., the bare surface (blue curves) has coercive field of around 1.5 mT (~30 times the Earth's field). When chiral crystals are present this coercivity increases to 2.5 mT (~50 times the Earth's field). We decided to highlight the added magnetic hardness due to the presence of chiral crystals without being too explicit about the bare coercivity of the material as this number is system dependent and likely to change from material to material (and with different thickness of the same material). The added coercivity by the chiral crystals is more physical and relevant.

9. *Looking at Fig. 2. C, we see that enantiopure RAO, formed elsewhere, can then recrystallize on a randomly magnetized domain and align the spins of that domain and create the avalanche magnetization process, which allows further RAO to crystallize in enantiopure form. However, the implication from the diagram is that when the boundary of that domain is reached, the avalanche magnetization process is halted. Is this correct? If so it should be stated, and if not I would change the diagram.*

Yes, we agree with the conclusion of the reviewer that the avalanche magnetization process is due to the finite range of magnetic domains. As a domain flips due to the chiral crystals, nearby domains are not readily affected by the chiral crystals they do not interact with. However, there is a caveat. As the domain that is in direct contact with the crystals flips and attracts more chiral molecules faster, inevitably it will attract chiral molecules at its magnetic domain boundary and into the nearby magnetic domains. Then another avalanche magnetization process gets triggered. So, the chirality induced magnetization is continuous and not halted BUT avalanche magnetization process only happens within the scale of the magnetic domains. We thank the reviewer for this insightful remark, and we placed here an illustration to explain the caveat. We made a change in the caption to clarify this.

10. *Some of the experimental detail given in the main text could be moved to the Experimental section or SI. Eg. The majority, if not all, of lines 272-275, 384-404 and 490-498 could be removed.*

We made the necessary revisions as suggested. We thank the reviewer for their interest in our manuscript.

We thank **Reviewer #4** for their interest in our manuscript and their encouraging feedback. We are pleased to say that their remarks and suggestion to experiment further with a different compound have been immensely helpful in improving our manuscript. We will respond to their remarks in the same order they put them.

1. *RAO is quite a special molecule. The generalization of this effect to other systems is not yet warranted. Have the authors experimented with other conglomerate systems to show that this is not a special case?*

As pointed out by the reviewer RAO is a special molecule for prebiotic chemistry due to its central role in the synthesis of ribonucleotides, however, as a chiral compound it is not a very special one. Previous CISS experiments have shown similar spin-controlled phenomena with various other compounds such as amino acids, peptides, and DNA, to name a few. Tassinari et al. (2019) showed that conglomerate forming amino acids like asparagine and glutamic acid strongly interact with magnetic nickel surfaces and achieved their spin-selective enantioseparation. Ben-Dor et al. (2017) showed that α -helix polyaniline molecules magnetize magnetic cobalt surfaces due to the spin-exchange interaction where the magnetization direction is determined the handedness of the molecules, very similar to our case. Many other experiments with different chemical compounds have verified that CISS phenomena is a general one, due to the spin-chirality coupling irrespective of the idiosyncratic chemical properties of the chiral molecules.

Having said that, we acknowledge the skepticism of the reviewer and included new MOKE measurements with L- α -helix polyaniline molecules adsorbed on the same magnetic nickel surfaces (Fig. S26). These new measurements, which we also referred in the main text, further verify that the chirality-induced magnetization phenomenon we study in the manuscript is not special to RAO but due to the spin-exchange interaction between chiral molecules and magnetic surfaces. We thank the reviewer for their constructive criticism, which significantly improved our manuscript.

2. *The authors state that “the emergence of homochirality at the stage of RAO allows for the propagation of homochirality through RNA to peptides and therefore to the entire prebiotic network.” RAO can only lead to nucleoside C, and the challenge of making other nucleosides in a prebiotically relevant way remains. Likewise, the extension to peptides, which is cited as the “cyanosulfidic network” of Sutherland, is not straightforward, in that these amino acids are produced in racemic form in that network. You’d need to get to RNA itself producing amino acids to make this claim.*

We acknowledge the confusion of the reviewer and thank them for pointing it out. We added a new paragraph at the end of the introduction detailing how chiral

information can propagate from RNA to peptides, which significantly improved our manuscript. In fact, we have written a new paper (<https://arxiv.org/abs/2306.01803>) on this important matter and cited it in the new paragraph.

“Last but not least, the emergence of homochirality at the stage of RAO allows for the efficient transfer of homochirality through RNA to peptides due to the high stereoselectivity in the attachment of L-amino acids to tRNA analogs composed of D-ribonucleotides. Therefore, the biological homochirality problem can be simplified by focusing on the production of a single shared RNA precursor with uniform chirality. Once homochirality is achieved in RAO, it can be effectively transferred from RNA to peptides, and eventually, through enantioselective catalysis, to metabolites—therefore to the entire prebiotic network. With these features, RAO is an important prebiotic compound which can play a central role in the emergence of biological homochirality.”

RAO is a precursor to the pyrimidines: cytidine and uracil, as uracil can be obtained as an UV irradiation product of cytidine. However, a prebiotically plausible synthesis of RNA purines (adenine and guanine) is still an open problem beyond their homochirality. We acknowledge this missing link in prebiotic chemistry and hope to find a route by which RNA purines can be synthesized from RAO as well.

3. *It is curious that the authors cite only a 1953 theoretical paper to discuss “persistent amplification” of an imbalance. The current work is not related to the mathematical model in the Frank paper. The authors have neglected important work in this area: Kondepudi, Blackmond, Videma, Soai.*

We acknowledge the criticism of the reviewer, and we added an entire paragraph reviewing the relevant literature, including Kondepudi, Blackmond, Viedma, and several others. We decided to omit Soai’s work as it has no relevance to prebiotic chemistry, although it is highly important for enantioselective autocatalysis. We thank the reviewer for their suggestion, which significantly improved our manuscript.

We thank **Reviewer #5** for their interest in our manuscript and their constructive criticism. We are pleased to say that their remarks and criticism have been appreciated and were helpful in clarifying our conceptual understanding of the discussed phenomena and our manuscript. We will respond to their remarks in the same order they put them.

“The manuscript by Ozturk et al. investigates the interactions between chiral molecules and the magnetic state of a sample, in particular with respect to the magnetic properties. The title of the manuscript may be flashy, but to my view does not adequately describe the real content of the paper. First, the studies do not only encompass magnetite, but also Ni films. Second, the main message of the manuscript is that the observed effects can all be traced back to chirality-induced spin selectivity (CISS). Other mechanisms are not discussed and leave the impression that CISS is the only mechanism. The claim is that the CISS effect establishes a strong coupling mechanism between electron spin and molecular chirality. There are many groups worldwide addressing this issue, but with mixed success. Therefore, given the controversial understanding and interpretation of CISS I find this view somewhat biased.”

It seems that this first paragraph is the key for the reviewer’s critical opinion on our manuscript, which we appreciated. We acknowledge the reviewer’s criticism, and we must state that their critical approach is much needed for a wider appreciation of the CISS effect. Hence, we would like to address it in detail.

Indeed, the CISS effect is the only effect observed, thus far, that relates the adsorption of chiral molecules to the sense of magnetization of a ferromagnetic substrate. In the current literature there is no alternative, room-temperature-robust phenomena, coupling molecular chirality to electron spin. The CISS effect is currently studied by many groups worldwide and the experimental results consistently verify that chiral molecules strongly interact with the electron spin due to the “spin-filtering effect of chiral molecules” established by the CISS effect (Ref 15 - Göhler et al. (2011), Ref 16 - Kettner et al. (2015)). Moreover, the reproducibility of the results and the large and robust signals observed are consistent through the published data.

As shown in our manuscript, and in many other previous reports (Ref 17 - Ben-Dor et al. (2017) and Ref 18 - Banarjee-Ghosh et al. (2018) to name a few), molecular chirality, not idiosyncratic chemical interactions, influence and interact with the surface spins. The fact that we do the control experiments with opposite enantiomers and get opposite spin-polarizations along the chiral molecular axis and do not get any spin-polarization with achiral molecules is a testament to the aforementioned phenomenon. Moreover, we experimented with another chiral molecule (alpha-helix polyalanine) that is well-studied in previous reports (Ref. 17 and 18) and decided to include these measurements in our manuscript as well

(Fig. S26). Once again, in a different setting with a different molecule, we observed a strong induced magnetization due to the molecular chirality of the chiral molecules that are covalently bonded to the magnetic substrate.

It is true and well-known that there is still a lack of agreement among theoreticians regarding the precise theoretical description of the effect. However, like many novel and surprising effects (such as giant magnetoresistance, high T_c superconductivity and others) the theoretical understanding is lagging behind the experimental observations and often, even behind the actual applications of those effects in consumer technologies.

Hence, based on the above, and based on the experimental observations in the present manuscript, it is reasonable to base the observations on the CISS effect, as there is no other alternative description.

“The samples investigated in this study are basically created by a liquid phase approach, bringing enantiopure solutions of D-RAO or L-RAO onto the surfaces and letting them dry to form crystals at the surface. The templates were magnetite crystals, magnetite thin films and Ni thin films. As the CISS arguments in the manuscript draw heavily on the exchange interaction between magnetic surface and the molecules, one immediate question arises: What is the chemical and electronic nature of the contact between the molecules and the magnetic surfaces? The drying process is certainly leaving behind some residue around the molecular crystals and very likely also at the interfaces. It is also known that oxides, such as magnetite, may have a different chemical composition at the surface, which deviates from the bulk one (if the oxygen concentration at the surface is changed, the material may turn from Fe_3O_4 to Fe_2O_3 , which is actually an antiferromagnet). The same is going to happen with magnetite films and the Ni films, which however have been protected by a Au cover layer (5nm). Therefore, in all of the cases, it cannot be assumed that the molecular crystals are in direct contact with a ferromagnet. As a consequence, electronic exchange interaction which requires an overlap of the participating wave functions will not work, but this is argued to be the main mechanism of CISS.”

RAO molecules are likely to interact with the iron-oxide surfaces through their highly polarizable diol moieties, in a non-covalent fashion. Magnetite surfaces are quite stable under ambient conditions and their oxidation to hematite is extremely slow at room temperature. We investigated the surface composition of our magnetite surfaces by XPS measurements and verified that the surface is purely magnetite down to the detection limit of XPS. Moreover, as we mentioned before, we added new measurements using alpha helix polyalanine molecules with thiol ends, through which the molecules covalently bind to the gold capping layer of

the Ni/Au substrate. HS-Au interactions is a very well-studied one and, in this system, too, we observe similar behavior.

Several previous studies found the correlation between the adsorption kinetics of chiral molecules and the sense of magnetic direction of ferromagnetic surfaces. It is important to realize that even if the ferromagnetic surface is coated with a monolayer of small molecules, water etc., because of the well-established proximity effect at the interface between the ferromagnet and the adsorbed layer, spin alignment takes place also in the adsorbed layer (Ref. 17, Vager et al. (2002), Carmeli et al. (2003)).

In past studies, ferromagnetic domains could be identified by their interaction with a tip of an AFM to which chiral molecules were attached (Ref 40 - Ziv et al. (2019)). This experiment proved that spin-exchange interaction indeed has an important contribution to the molecule-ferromagnet interaction. The importance of spin-exchange was also proven for interaction between chiral molecules and ferromagnetic surface covered with small achiral molecules (Ref 41 - Kapon et al. (2021)).

*“The magnetic data, particularly, the domain imaging experiments look interesting. Apparently, there is an influence of the molecular crystals (D-RAO) on the domain formation and the magnetization behavior of the Ni/Au film structure. Around the line-shaped trace of the D-RAO crystals, the coercivity is enlarged. **Can the authors exclude that this is just a consequence of the changed chemistry at the surface/chemistry due to preparation procedure of the molecular crystals?** The enlarged coercivity is going to change the magnetic domain structure, particularly, if the max field values are chosen between the switching fields of the D-RAO area and the Ni/Au area. Again, I believe that the distance argument (see 1) eliminates CISS as a likely mechanism for this behavior.”*

It is well established in spintronics that a thin layer of gold does reflect the spin property of the underlying ferromagnetic layer. This issue was established in the literature (Ref 48 - Mondal et al. (2016) and Zhu et al. Multistate Switching of Spin Selectivity in Electron Transport through Light-Driven Molecular Motors (2021) and references therein) and now we refer to it also in the manuscript. We thank the reviewer for allowing us to clarify this.

Yes, we excluded this possibility by doing control experiments. With achiral molecules like glycine and NaCl crystals, we did not observe a change in the magnetic properties of the surface when the experiments are repeated with achiral molecules (Extended Data Fig 2). Moreover, the magnetization reversal behavior with L- and D- RAO on Fe₃O₄, as shown by the CD measurements,

indicates a coupling of the molecular chirality to the surface spins in an enantiospecific manner (Fig. 3b). Both for L-RAO and D-RAO the nature of the chemical contact with the surface is identical. However, the difference is in the change of the asymmetry of the chiral potential influencing the surface spins. Furthermore, experiments with other chiral molecules, like L-alpha helix polyaniline, that is covalently bonded to gold, shows a similar behavior to chiral RAO molecules (Fig. S26). The nature of the chemical contact between RAO and alpha helix polyaniline with the magnetic surface is completely different. Yet, both molecules are chiral and therefore they both lead to a spin-polarization along their chiral molecular axes due to the CISS effect and interact with the magnetic substrate.

“The AFM investigations are interesting, too, but again misleading. The authors talk about the spin-polarization of the RAO molecules, but their experiment never measures a spin-polarization. What they are actually measuring are differences in the I(V) curves by means of a non-magnetic (!) tip as a function of the applied magnetic field direction. Both L-RAO and D-RAO samples indeed exhibit opposite changes with the magnetic field direction. This indicates a magnetic field effect at work in the I(V) curves, but is not a measurement of the spin-polarization. Again, one has to keep in mind that the Ni layer is protected by the 5 nm Au film and the Au/molecule surface has been treated by spin coating. As a result the average distance between Ni and molecules is larger than 5 nm. I don't understand how the exchange interaction claimed for CISS is going to bridge this large distance. In fact, even the authors state that the exchange interaction may be strong, but is very short ranged.”

The reviewer is right that the magnetic effect on the I-V curves is not direct evidence for spin polarization, but rather it can be a result of orbital angular momentum selection rules. Strictly speaking, spin is not a good quantum number for electrons moving within the chiral potential, if only due to the spin orbit coupling. However, there is evidence based on Hall (Eckshtain-Levi et al., *Nature Comm.* (2016), Kumar et al., *PNAS* (2017)) and Hanle effect (Lu, Wang, and Xiao et al., *Science Advances* (2019)) measurements that directly relate the electron transport through chiral molecules to spin selectivity. This is supported by photoemission studies that directly detect the longitudinal spin polarization (Ref. 15 and 16) and show excellent correlation between the “spin-polarization” derived from the I-V curves and the that measured using Mott detector (Ref. 15 and 16)

“These are my major difficulties with this manuscript. Although it shows some interesting observations, which still await interpretation, I have strong doubts that these can be unanimously explained with the CISS effect. Therefore, I cannot recommend this paper for publication in Nat. Comm.”

To summarize, we recognize and value the reviewer's critical perspective regarding the CISS experiments. It is the community's and our responsibility to experiment with various systems, repeat the experiments as much as need and do multiple controls to establish the coupling we claim to exist. In this manuscript, we have repeated our results and have done multiple controls (Extended Data Fig.1 and 2, Fig S16) and experimented with various other systems (Fig. S25, Fig. S26) to establish the validity of our claims. We strongly believe that our data is sound and repeatable, and our explanation is the only plausible one, supported by its self-consistency with previous research.

We acknowledge the skepticism of the reviewer to account for our observations with the CISS effect, which we partially attribute to the lack of full theoretical understanding of the phenomena from the first principles. Despite the difficulty to provide *ab initio* calculations of the effect, which is an ongoing research, previous research describing the experimental results related to ours must serve as evidence of its credibility. As previously stated, several effects in physics have taken many years to be explained or remain unexplained, such as high T_c superconductivity.

Taking this into account, we are optimistic that the reviewer, regardless of any skepticism they may have, can view our work as presenting intriguing new findings significant for the origins of life research and suggesting an explanation that aligns with current knowledge. We sincerely thank the reviewer for enabling this engaging discussion.

Reviewers' Comments:

Reviewer #1:

Remarks to the Author:

The authors have responded to my initial thoughts and issues satisfactorily. I recommend publication of the manuscript.

Reviewer #2:

Remarks to the Author:

Authors carefully addressed questions raised in the first round of review. The manuscript can be recommended for publication.

Reviewer #3:

The manuscript has been improved but there still remain a few minor points to be resolved, for brevity I copied the last round of review and add new comments in purple.

We thank **Reviewer #3** for their interest in our manuscript and their encouraging feedback. We are pleased to say that their remarks have been helpful in improving our manuscript. We will respond to their remarks in the same order they put them.

1. *Abstract, lines 13-14: An environmental factor is not necessarily required to achieve homochirality, as demonstrated by the Soai reaction. Whilst a prebiotic example of this type of auto amplification of chirality has not been demonstrated yet, in theory it could be possible. Please amend.*

At the end of the mentioned sentence, “required” was changed to “desirable”. Although for breaking the symmetry, spontaneous chemical amplification is possible, in order to maintain the imbalance persistently an environmental factor is highly beneficial. This ensures that at various stages of prebiotic chemistry, synthesis of chiral molecules can be along a common direction biased by the environmental agent. We thank the reviewer for highlighting this.

Changing one word in the abstract has not resolved this point, in fact, it is more enigmatic than before. The authors seem to be suggesting that if an environmental factor could be used as chiral agent, there would be a persistence of chiral selection for 1 enantiomer/diastereomer over another i.e. if the chemistry stalled and then started again at a later time, or was occurring in 2 or more locations in the same environment, the same chiral bias would be observed, which would greatly assist prebiotic chemistry. Whereas if a chemical reaction provided an auto amplification of chirality, this would not necessarily be the case. If this is what the authors mean, I agree, but they need to alter the abstract so this concept is expressed clearly.

2. *Abstract, lines 16-17: Following on from above, please correct e.g. “..magnetic surfaces is crucial for this approach..”*

In this sentence we do not mean that “magnetic surfaces” are universally crucial to break the chiral symmetry. What we mean is simply that due to our previous success in utilizing magnetized surfaces as chiral agents, inducing a net magnetization is crucial. This is implied in the previous sentence.

I appreciate the explanation and clearly the authors know what they mean, but I don't believe they communicate it very well. I would suggest something along the following lines

“Thus, inducing net magnetization on magnetic surfaces is crucial for this approach and can be achieved by Earth's magnetic field. Here, we report the

avalanche magnetization of magnetite (Fe₃O₄) by crystallizing enantiopure RAO, which induces uniform, rather than net, magnetization through the CISS effect. Chirality-induced magnetization.....”

3. *1st paragraph, lines 26-27: Obtaining high yields of chemical reactions doesn't necessarily require enantiopure compounds, the emphasis must be on obtaining functional polymers, which is well known.*

We changed the sentence to: “Reaching and maintaining homochirality is crucial for a robust prebiotic network **efficiently** producing functional polymers.” We thank the reviewer for the suggestion.

Acknowledged

4. *4th paragraph, lines 60-62: Considering the length of the manuscript, another sentence or two describing briefly how RNA can (potentially) confer chirality to peptides won't go amiss, particularly as this appears to be one of the most salient points of the paper.*

We thank the reviewer for their suggestion which we totally agree with. We made the following changes and cited our most recent paper on how homochirality can propagate in a prebiotic network. We also separated this paragraph to isolate and highlight its important content.

“Last but not least, the emergence of homochirality at the stage of RAO allows for the efficient transfer of homochirality through RNA to peptides due to the high stereoselectivity in the attachment of L-amino acids to tRNA analogs composed of D-ribonucleotides. Therefore, the biological homochirality problem can be simplified by focusing on the production of a single shared RNA precursor with uniform chirality. Once homochirality is achieved in RAO, it can be effectively transferred from RNA to peptides, and eventually, through enantioselective catalysis, to metabolites—therefore to the entire prebiotic network. With these features, RAO is an important prebiotic compound which can play a central role in the emergence of biological homochirality.”

Substantially improved

5. *Fig. 1. I'm not sure this scheme makes sense. Once avalanche magnetization has taken place (top left) the arrow points to a magnetite surface with randomly magnetized domains and RAO crystallizing on it. Shouldn't the arrow point to a magnetite surface with a uniform surface magnetization and racemic RAO crystallizing onto it in enantiopure form i.e. the need for further recrystallisation (arrow from bottom right to bottom left) is not required to achieve high ee. Also, for the bottom left image, should the text in black not read "nearly enantiopure RAO"?*
caption

We acknowledge the confusion of the reviewer and thank them for pointing it out; however, we illustrated the magnetic substrate as 2D cross sections from the side (front view), and the relevant part of the substrate is its top surface. We decided to do so to display magnetic domains and emphasize that the chirality-induced magnetization phenomenon is a surface effect. Because the illustrations are 2D cross sections, it is not possible to draw the top-left arrow to the very top of magnetic substrate. However, we made clarifying changes in the figure by moving the text to the top and making the “higher selectivity” arrow start from the top. And we also clarified that in Fig. 2., we illustrated the surface as a projection from the top (top view). We changed the text to “nearly enantiopure RAO”. We thank the reviewer for these suggestions.

I'm not what the authors mean with this response. Maybe I didn't express myself well enough. This diagram is supposed to show racemic RAO crystallizing in enantioenriched form onto magnetite which has a net magnetization. Recrystallizing this RAO again gives (more or less) enantiopure RAO, and this drives uniform magnetization of the magnetite. The diagram serves well to this point.

Adding a reaction arrow labelled 'higher selectivity' is not helpful. Do the authors mean that the enantiopure RAO formed (top left image) can now crystallize on

other magnetite and induce uniform magnetization, or do they mean that, now the magnetite has uniform magnetization, further racemic RAO crystallizing on it will be enantiopure? I believe it's the latter case, but either way, the image (bottom right) does not depict either of these events. The authors need to make some effort and actually change the Figure. I would think removing the higher selectivity arrow and putting a new arrow from the top left image going left to a new image will be the way to solve the problem e.g. reaction arrow with racemic RAO beside it going to the same uniform magnetized surface but with enantiopure RAO crystals on it.

6. *Fig 3. line 263: "..D-RAO crystals (thick blue curve).."*

We are afraid we do not understand the confusion here. D-RAO crystals correspond to the thick red curve and L-RAO crystals correspond to the thick blue curve in the panel of Fig.3b.

This is a simple typo, should be "L-RAO crystals (thick blue curve) give negative and D-RAO crystals (thick red curve) give positive...."

7. *How big are the naturally occurring domains of magnetized Fe₃O₄? The range of typical sizes would be of interest to the reader and help visualize the process.*

We thank the reviewer for their intriguing question that we are currently investigating for a future work. The domain structure and hence the magnetic properties of natural magnetite is heavily dependent on how it forms in the environment. If magnetite crystallizes from a more concentrated solution of ferrous iron, typically the magnetite crystals become smaller (10-50 nm) and have single-domain structure. If magnetite crystals are larger, they can either be in the pseudo-single domain regime or in the multi-domain regime. The magnetite we are currently investigating is in the pseudo-single regime with a crystal size of around 100 nm. These magnetite crystals are ideal for high surface spin-polarization without being magnetically too soft. As they exist in the pseudo-single domain regime, each crystal can be thought of as a purely spin-polarized unit. On the other hand, the synthetic magnetite substrates we fabricated for the experiment have much larger domains, but, still not resolvable by optical imaging. We estimate their range to be around and less than 1 micron. That's why for the MOKE experiments we employed Ni/Au surfaces with larger domains such that we can 'watch' the domain flipping. Magnetic domains of the Ni/Au substrate before the RAO crystals are adsorbed can be seen in Fig. S19. They are around 10 microns.

If this is to be published at a later date it need not be included now.

8. *From the MOKE experiments, it is impressive to see the resistance to spin flipping when chiral crystals are present. When the crystals are washed away, however, presumably the magnetic hardness of that area is lost? If so, how does the magnetic resistance compare to that of Earth's magnetic field? We are told in the presence of RAO it is ~ 20 times.*

When the crystals are washed away, the added magnetic hardness to the surface is lost. However, the bare magnetic hardness of the surface still remains. As seen in Fig. 4c., the bare surface (blue curves) has coercive field of around 1.5 mT (~30 times the Earth's field). When chiral crystals are present this coercivity increases to 2.5 mT (~50 times the Earth's field). We decided to highlight the added magnetic hardness due to the presence of chiral crystals without being too explicit about the bare coercivity of the material as this number is system dependent and likely to change from material to material (and with different thickness of the same material). The added coercivity by the chiral crystals is more physical and relevant.

Apologies, my mistake ~50 times.

I think it is important to include some discussion of this in the paper as we cannot expect RAO to remain bound to magnetite forever, and, if anything, strengthens the authors argument regarding persistence of an environmental chiral agent i.e. even if the initial RAO is lost, the next time it crystallizes it will be enantiopure and same enantiomer. Of course, the reader needs to be made aware of the uncertainties, but even if its an order of magnitude out the direction of magnetization will be preserved.

9. *Looking at Fig. 2. C, we see that enantiopure RAO, formed elsewhere, can then recrystallize on a randomly magnetized domain and align the spins of that domain and create the avalanche magnetization process, which allows further RAO to crystallize in enantiopure form. However, the implication from the diagram is that when the boundary of that domain is reached, the avalanche magnetization process is halted. Is this correct? If so it should be stated, and if not I would change the diagram.*

Yes, we agree with the conclusion of the reviewer that the avalanche magnetization process is due to the finite range of magnetic domains. As a domain flips due to the chiral crystals, nearby domains are not readily affected by the chiral crystals they do not interact with. However, there is a caveat. As the domain that is in direct contact with the crystals flips and attracts more chiral molecules faster, inevitably it will attract chiral molecules at its magnetic domain boundary and into the nearby magnetic domains. Then another avalanche magnetization process gets triggered. So, the chirality induced magnetization is continuous and not halted BUT avalanche magnetization process only happens within the scale of the magnetic domains. We thank the reviewer for this insightful remark, and we placed here an illustration to explain the caveat. We made a change in the caption to clarify this.

This looks better

10. *Some of the experimental detail given in the main text could be moved to the Experimental section or SI. Eg. The majority, if not all, of lines 272-275, 384-404 and 490-498 could be removed.*

We made the necessary revisions as suggested. We thank the reviewer for their interest in our manuscript.

Reviewer #4:

Remarks to the Author:

The authors give this lengthy explanation about the problem with the purines:

"RAO is a precursor to the pyrimidines: cytidine and uracil, as uracil can be obtained as an UV irradiation product of cytidine. However, a prebiotically plausible synthesis of RNA purines (adenine and guanine) is still an open problem beyond their homochirality. We acknowledge this missing link in prebiotic chemistry and hope to find a route by which RNA purines can be synthesized from RAO as well."

However, this discussion must be part of the manuscript, not just the review. The word purine does not appear in the manuscript. This peer review exercise is not simply a dialog between reviewers and authors. Other readers will have the same question. The authors have failed to address the question that RAO leads only to half of the letters of RNA. In this context, The paragraph copied below is misleading:

"Last but not least, the emergence of homochirality at the stage of RAO allows for the efficient transfer of homochirality through RNA to peptides due to the high stereoselectivity in the attachment of L-amino acids to tRNA analogs composed of D-ribonucleotides. Therefore, the biological homochirality problem can be simplified by focusing on the production of a single shared RNA precursor with uniform chirality. Once homochirality is achieved in RAO, it can be effectively transferred from RNA to peptides, and eventually, through enantioselective catalysis, to metabolites—therefore to the entire prebiotic network. With these features, RAO is an important prebiotic compound which can play a central role in the emergence of biological homochirality."

I repeat my concern that over-hyping the issue takes away from the very nice results.

Reviewer #5:

Remarks to the Author:

no further comments

We thank **Reviewer #3** for their feedback and remarks. Their meticulous examination of our manuscript has greatly enhanced the quality of our paper. We have addressed the remaining aspects they raised in the order they presented them.

1. *Changing one word in the abstract has not resolved this point, in fact, it is more enigmatic than before. The authors seem to be suggesting that if an environmental factor could be used as chiral agent, there would be a persistence of chiral selection for 1 enantiomer/diastereomer over another i.e. if the chemistry stalled and then started again at a later time, or was occurring in 2 or more locations in the same environment, the same chiral bias would be observed, which would greatly assist prebiotic chemistry. Whereas if a chemical reaction provided an auto amplification of chirality, this would not necessarily be the case. If this is what the authors mean, I agree, but they need to alter the abstract so this concept is expressed clearly.*

Yes, this is precisely what we meant. In fact, we discussed this matter at length in the discussion section under the subsection “Implications on prebiotic chemistry”. To clarify our point in the abstract we made the following change:

“... To achieve and maintain homochirality within a prebiotic network, the presence of an environmental factor acting as a chiral agent and providing a persistent chiral bias to prebiotic chemistry is highly advantageous. ... ”

2. *I appreciate the explanation and clearly the authors know what they mean, but I don't believe they communicate it very well. I would suggest something along the following lines:*

“Thus, inducing net magnetization on magnetic surfaces is crucial for this approach and can be achieved by Earth's magnetic field. Here, we report the avalanche magnetization of magnetite (Fe₃O₄) by crystallizing enantiopure RAO, which induces uniform, rather than net, magnetization through the CISS effect. Chirality-induced magnetization ...”

We acknowledge the reviewer's confusion and thank them for their suggestion, which significantly improved our abstract. We made the following revision:

“Magnetized surfaces are prebiotically plausible chiral agents due to the chiral-induced spin selectivity (CISS) effect, and they were utilized to attain homochiral ribose-aminooxazoline (RAO), an RNA precursor. However, natural magnetic minerals are typically weakly magnetized, necessitating mechanisms to enhance their magnetization for their use as effective chiral agents. Here, we report the

magnetization of magnetic surfaces by crystallizing enantiopure RAO, whereby chiral molecules induce a uniform surface magnetization due to CISS effect, which spreads across the magnetic surface akin to an avalanche.”

5. *I'm not what the authors mean with this response. Maybe I didn't express myself well enough. This diagram is supposed to show racemic RAO crystallizing in enantioenriched form onto magnetite which has a net magnetization. Recrystallizing this RAO again gives (more or less) enantiopure RAO, and this drives uniform magnetization of the magnetite. The diagram serves well to this point. Adding a reaction arrow labelled 'higher selectivity' is not helpful. Do the authors mean that the enantiopure RAO formed (top left image) can now crystallize on other magnetite and induce uniform magnetization, or do they mean that, now the magnetite has uniform magnetization, further racemic RAO crystallizing on it will be enantiopure? I believe it's the latter case, but either way, the image (bottom right) does not depict either of these events. The authors need to make some effort and actually change the Figure. I would think removing the higher selectivity arrow and putting a new arrow from the top left image going left to a new image will be the way to solve the problem e.g. reaction arrow with racemic RAO beside it going to the same uniform magnetized surface but with enantiopure RAO crystals on it.*

We acknowledge the reviewer's confusion and appreciate their efforts to improve our manuscript. We meant along the lines of the latter, as the reviewer correctly guessed. Once magnetite is uniformly magnetized subsequent RAO crystallizations from its incoming racemic solution will be enantiopure. So, the “higher selectivity” arrow is just the one that closes the feedback loop: second run/loop will be more enantioselective than the first one, third run/loop will be more selective than the second one, so on and so forth until both the surface is uniformly magnetized, and the crystals are enantiopure. So, it is not a real arrow, but it implies that the uniformly magnetized surface will be re-used as a template for crystallization, but now with higher selectivity.

To clarify, we replaced the “higher selectivity” marker with “feedback” and replaced the solid arrow with a dashed one. We want to emphasize the presence of closed feedback, so we did not want to totally remove the arrow. We also revised the caption and tried to clarify our point:

“...Hence, as enantiopure crystals of RAO form on the magnetic surface, the chirality-induced magnetization allows for obtaining surfaces with uniform magnetization. These uniformly magnetized surfaces can be employed again for successive crystallizations of the incoming racemic material (dashed arrow), with significantly enhanced enantioselectivity compared to surfaces magnetized under the geomagnetic field.”

6. We fixed the typo.

8. *I think it is important to include some discussion of this in the paper as we cannot expect RAO to remain bound to magnetite forever, and, if anything, strengthens the authors argument regarding persistence of an environmental chiral agent i.e., even if the initial RAO is lost, the next time it crystallizes it will be enantiopure and same enantiomer. Of course, the reader needs to be made aware of the uncertainties, but even if it's an order of magnitude out the direction of magnetization will be preserved.*

We thank the reviewer for this suggestion, and we agree that adding such a discussion is valuable. We added the following to our discussion under "Implications on prebiotic chemistry":

"... Finally, it increases the magnetic resistance (coercivity) of the surface by about 20 times the modern geomagnetic field. Of course, this added coercivity due to the chiral molecules will diminish as the molecules are washed away. Nevertheless, the chirality-induced surface magnetization will persist even after the molecules are removed and it will be protected by the bare coercivity of the magnetic surface, which is about 30 times the modern geomagnetic field for our surfaces."

We thank **Reviewer #4** for their remarks, and we acknowledge their constructive criticism. According to their suggestions, we made the necessary revisions, which improved the quality of our paper.

The authors give this lengthy explanation about the problem with the purines:

“RAO is a precursor to the pyrimidines: cytidine and uracil, as uracil can be obtained as an UV irradiation product of cytidine. However, a prebiotically plausible synthesis of RNA purines (adenine and guanine) is still an open problem beyond their homochirality. We acknowledge this missing link in prebiotic chemistry and hope to find a route by which RNA purines can be synthesized from RAO as well.”

However, this discussion must be part of the manuscript, not just the review. The word purine does not appear in the manuscript. This peer review exercise is not simply a dialog between reviewers and authors. Other readers will have the same question. The authors have failed to address the question that RAO leads only to half of the letters of RNA. In this context, the paragraph copied below is misleading: [last paragraph of the introduction]

According to the reviewers' remarks we made significant changes to the last paragraph of our introduction and included our previous discussion with the reviewer here. Here is the revised paragraph:

“Last but not least, the emergence of homochirality at the stage of RAO can allow for the efficient transfer of homochirality through RNA to peptides due to the high stereoselectivity in the attachment of L-amino acids to tRNA analogs composed of D-ribonucleotides. However, it is important to acknowledge that, based on current evidence, RAO serves as a precursor solely to RNA pyrimidines. The prebiotically plausible synthesis of purine ribonucleotides, however, remains an unresolved challenge. If a route to RNA purines from RAO is also discovered, the biological homochirality problem can be simplified by focusing on the production of a single shared RNA precursor with uniform chirality. In such a scenario, once homochirality is achieved in RAO, it can be effectively transferred from RNA to peptides, and eventually, through enantioselective catalysis, to metabolites—therefore to the entire prebiotic network. With these features, RAO is a promising prebiotic compound which can play a central role in the emergence of biological homochirality.”

Reviewers' Comments:

Reviewer #3:

Remarks to the Author:

There is one point that still remains, regarding Figure 1.

I realise the authors want to show there is a feedback mechanism that can operate, but the problem is that their figure does not convey what they wish to express. If the authors really wish to minimise their efforts, I suggest an alteration of the diagram as shown in the attached.

stereoselective and enantioselective crystallization

and P AO can amplify the natural magnetization.

Reviewer #4:

Remarks to the Author:

the authors have adequately addressed my concerns.

We thank **Reviewer #3** for their feedback. Their meticulous examination of our manuscript has greatly enhanced the quality of our paper.

“I realise the authors want to show there is a feedback mechanism that can operate, but the problem is that their figure does not convey what they wish to express. If the authors really wish to minimise their efforts, I suggest an alteration of the diagram as shown in the attached.”

We have implemented the recommended modification in Figure 1, and the updated illustration is presented below. We rotated the figure 90 degrees, positioning the uniformly magnetized surface beneath the racemic molecules, and then neatly incorporated the suggested arrow to connect them. We thank the reviewer for their recommendation.

Fig. 1. A cooperative feedback between the magnetic surface and RAO can amplify the natural magnetization. Authigenic magnetic minerals get magnetized while they form under a geomagnetic field. Sedimentary rock surfaces with these magnetic inclusions carry a net remanent magnetization. This natural net magnetization is not uniform albeit statistically significant over the scale of a hemisphere and can allow for spin-selective asymmetric processes due to the CISS effect. In our previous work [3], we have shown that an essential RNA precursor, ribose-aminooxazoline (RAO), can selectively crystallize from a racemic mixture of pentose aminooxazolines on a magnetized magnetite (Fe₃O₄) surface (1). Although, with a subsequent re-crystallization, we obtained nearly enantiopure crystals of RAO (2); in a natural environment, on a surface with non-uniform magnetization, this process will inevitably be less selective. However, the interaction between the magnetic surface and chiral molecules is reciprocal: chiral molecules can magnetize magnetic surfaces due to the spin-exchange and magnetic dipolar interactions. Hence, as enantiopure crystals of RAO form on the magnetic surface, the *chirality-induced avalanche magnetization* allows for obtaining surfaces with uniform magnetization (3). These uniformly magnetized surfaces can be employed again for successive crystallizations of the incoming racemic material (red arrow), with significantly enhanced enantioselectivity compared to surfaces magnetized under the geomagnetic field. Therefore, the self-reinforcing cooperative feedback between the magnetic surface and chiral molecules can enhance the natural magnetization and set the stage for highly selective asymmetric processes on early Earth.